# GRAFFE: GRAPH REPRESENTATION LEARNING ENABLED VIA DIFFUSION PROBABILISTIC MODELS

## ABSTRACT

Diffusion probabilistic models (DPMs), widely recognized for their potential to generate high-quality samples, tend to go unnoticed in representation learning. While recent progress has highlighted their potential for capturing visual semantics, adapting DPMs to graph representation learning remains in its infancy. In this paper, we introduce **Graffe**, a self-supervised diffusion model proposed for graph representation learning. It features a graph encoder that distills a source graph into a compact representation, which, in turn, serves as the condition to guide the denoising process of the diffusion decoder. To evaluate the effectiveness of our model, we first explore the theoretical foundations of applying diffusion models to representation learning, proving that the denoising objective implicitly maximizes the conditional mutual information between data and its representation. Specifically, we prove that the negative logarithm of denoising score matching loss is a tractable lower bound for the conditional mutual information. Empirically, **Graffe** delivers competitive results under the linear probing setting on node and graph classification, achieving state-of-the-art performance on 9 of the 11 real-world datasets. These findings indicate that powerful generative models, especially diffusion models, serve as an effective tool for graph representation learning.

## 1 INTRODUCTION

Self-supervised learning (SSL), which enables effective data understanding without laborious human annotations, is emerging as a key paradigm for addressing both generative and discriminative tasks. When we revisit the evolution of SSL across these two tasks, interestingly, a mutually reinforcing manner becomes evident: Progress in one aspect often stimulates progress in the other. For instance, autoencoder (Hinton & Salakhutdinov, 2006), which initially made a mark in feature extraction, laid the foundation for the success of VAEs (Kingma, 2013) for sample generation. Conversely, breakthroughs in generative tasks like autoregression (Radford, 2018) and adversarial training (Goodfellow et al., 2020), have deepened our understanding of representation learning, driving the development of iGPT (Chen et al., 2020) and BigBiGAN (Donahue & Simonyan, 2019).

Recently, diffusion models (Ho et al., 2020; Song et al., 2020) have demonstrated astonishing generation quality in different domains, particularly in terms of realism, detail depiction, and distribution coverage. A natural question arises: *can we draw on the successful experiences of diffusion models to enhance representation learning?* This issue is particularly pressing in the context of graph learning, since generation—the ability to create—plays a less critical role compared to discrimination on graphs, e.g., social networks, citation networks, and recommendation networks. The question seems not difficult to address, as generation is considered one of the highest manifestations of learning thus having powerful capability to learn high-quality representation (Krathwohl, 2002; Johnson et al., 2018; Wang et al., 2023; Hudson et al., 2024); however, the reality is much more complex.

To generalize the representation learning power of diffusion models on graph data, two main impediments must be addressed: ① **the non-Euclidean nature of graph data**, which complicates the direct application of diffusion models and necessitates consideration of both structural and feature information; ② **the absence of an encoder component in diffusion model** prevents us from obtaining explicit data representation and finetuning encoder in downstream tasks. Motivated to overcome these challenges, we investigate how to adapt diffusion models to graph representation learning and enhance their discrimination performance.

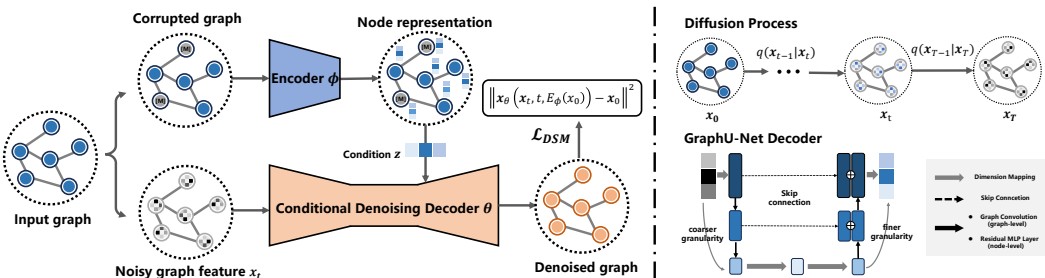

Figure 1: The overall framework of **Graffe**. **(Left)** The input graph has certain nodes corrupted and is subsequently fed into a GNN encoder to obtain node representations as the condition. The decoder then receives both the noisy graph features $\mathbf{x}_t$ and the condition $\mathbf{z}$ as inputs to perform denoising, aiming to restore the original node features $\mathbf{x}_0$. **(Right)** The diffusion process of graph features and the architecture of GraphU-Net decoder.

This work is particularly relevant to approaches that use diffusion models to capture high-level semantics for classification tasks while enhancing representational capacity. Those approaches can be broadly categorized into two main groups: *(i)* one treats part of the diffusion model itself as a feature extractor (*implicit-encoder pattern*) (Xiang et al., 2023; Chen et al., 2024; Yang et al., 2024). They obtain the latent representation from a certain intermediate layer, which inevitably exposes them to challenge ②. *(ii)* Another line of work jointly trains the diffusion model and an additional feature extractor (*explicit-encoder pattern*) (Abstreiter et al., 2021; Wang et al., 2023; Hudson et al., 2024). However, the latter pattern have struggled to surpass their contrastive and auto-encoding counterparts.

In this paper, we propose **Graffe**, which shares a philosophy similar to the explicit-encoder pattern. Starting with the optimization objective for diffusion-based SSL, we analyze diffusion representation learning (DRL) and show that it maximizes the mutual information lower bound between the learned representation and the original input, with more informative representations leading to lower denoising score matching loss, and vice versa. This suggests that DRL implicitly follows a principle akin to the InfoMax principle (Linsker, 1988; Hjelm et al., 2018), which we call the Diff-InfoMax principle. Furthermore, we observe from the frequency domain of graph features that DRL excels in capturing high-frequency information. Inspired by our theoretical insights, we instantiate our model with a graph neural network (GNN) encoder for explicit representation extraction and a tailored diffusion decoder, both trained from scratch in tandem. The encoder transforms the graph structure and feature information into a compact representation, which acts as a condition for the decoder together with noisy features to guide the denoising process. The main contributions of this work are three-fold:

❶ We theoretically prove that the negative logarithm of the denoising score matching loss is a tractable lower bound for conditional mutual information. Building on this, we introduce the Diff-InfoMax principle, an extension of the standard InfoMax principle, showing that DRL implicitly follows it.

❷ We propose an effective diffusion-based representation learning method catering to graph tasks, termed as **Graffe**. Equipped with random node masking and customized diffusion architecture for different task types, it can achieve sufficient graph understanding and obtain representations with rich semantic information.

❸ We conduct extensive experiments on 11 classification tasks under the linear protocol, spanning node- and graph-level tasks of diverse domains. Our method can achieve state-of-the-art or near-optimal performance across all datasets. On `Computer`, `Photo`, and `COLLAB` datasets, our model set a new accuracy record of 91.3%, 94.2% and 81.3%, respectively.

## 2 PRELIMINARY

### 2.1 BACKGOUND ON DIFFUSION MODEL

Diffusion Probabilistic Models (DPMs) construct noisy data through the stochastic differential equation (SDE):

$$\mathrm{d}\mathbf{x}_t = f(t)\mathbf{x}_t\mathrm{d}t + g(t)\mathrm{d}\boldsymbol{w}_t, \tag{1}$$

where $f(t), g(t) : \mathbb{R} \to \mathbb{R}$ is scalar functions such that for each time $t \in [0, T]$, $\mathbf{x}_t | \mathbf{x}_0 \sim \mathcal{N}(\alpha_t \mathbf{x}_0, \sigma_t^2 \mathbf{I})$, $\alpha_t$, $\sigma_t$ are determined by $f(t), g(t)$, $\boldsymbol{w}_t \in \mathbb{R}^d$ represents the standard Wiener process. Anderson (1982) demonstrates that the forward process (1) has an equivalent reverse-time diffusion process (from $T$ to 0) as the following equation so that the generating process can be equivalent to numerically solve the reverse SDE (Ho et al., 2020; Song et al., 2020).

$$\mathrm{d}\mathbf{x}_t = \left[ f(t)\mathbf{x}_t - g^2(t)\nabla_{\mathbf{x}} \log p_t(\mathbf{x}_t) \right] \mathrm{d}t + g(t)\mathrm{d}\bar{\boldsymbol{w}}_t, \qquad \mathbf{x}_T \sim p_T(\mathbf{x}_T), \tag{2}$$

where $\bar{\boldsymbol{w}}_t$ represents the Wiener process in reverse time, and $\nabla_{\mathbf{x}} \log p_t(\mathbf{x})$ is the score function. To get the *score function* $\nabla_{\mathbf{x}} \log p_t(\mathbf{x}_t)$ in (2), we usually take neural network $\boldsymbol{s}_{\boldsymbol{\theta}}(\mathbf{x}, t)$ parameterized by $\boldsymbol{\theta}$ to approximate it by optimizing the Denoising Score Matching loss (Song et al., 2020):

$$\boldsymbol{\theta}^* = \operatorname*{argmin}_{\boldsymbol{\theta}} \mathcal{L}_{DSM} = \operatorname*{argmin}_{\boldsymbol{\theta}} \mathbb{E}_t \left\{ \tilde{\lambda}(t) \mathbb{E}_{\mathbf{x}_0} \mathbb{E}_{\mathbf{x}_t | \mathbf{x}_0} \left[ \left\| \boldsymbol{s}_{\boldsymbol{\theta}}(\mathbf{x}, t) - \nabla_{\mathbf{x}_t} \log p_{0t}(\mathbf{x}_t | \mathbf{x}_0) \right\|_2^2 \right] \right\}, \tag{3}$$

where $\tilde{\lambda}(t)$ is a loss weighting function over time. In practice, several methods are used to reparameterize the score-based model. The most popular approach (Ho et al., 2020) utilizes a *noise prediction model* such that $\epsilon_{\boldsymbol{\theta}}(\mathbf{x}_t, t) = -\sigma_t \boldsymbol{s}_{\boldsymbol{\theta}}(\mathbf{x}_t, t)$, while others employ a *data prediction model*, represented by $\mathbf{x}_{\boldsymbol{\theta}}(\mathbf{x}_t, t) = (\mathbf{x}_t - \sigma_t \epsilon_{\boldsymbol{\theta}}(\mathbf{x}_t, t))/\alpha_t$. The DSM loss is equivalent to the following data prediction loss after changing the weighting function:

$$\mathcal{L}_{\mathbf{x}_0, DSM} = \mathbb{E}_t \left\{ \lambda(t) \mathbb{E}_{\mathbf{x}_0} \mathbb{E}_{\mathbf{x}_t | \mathbf{x}_0} \left[ \| \mathbf{x}_\theta(\mathbf{x}_t, t) - \mathbf{x}_0 \|^2 \right] \right\}. \tag{4}$$

## 2.2 INFOMAX PRINCIPLE

Unsupervised representation learning is a key challenge in machine learning, and recently, there has been a resurgence of methods motivated by the InfoMax principle (Linsker, 1988; Hjelm et al., 2018). Mutual Information (MI) quantifies the "amount of information" obtained about one random variable $X$ by observing the other random variable $Y$. Formally, the MI between $X$ and $Y$ with joint density $p(x, y)$ and marginal densities $p(x)$ and $p(y)$, is defined as the Kullback-Leibler divergence between the joint distribution and the product of the marginal distribution

$$I(X; Y) = D_{KL}(P_{(X,Y)} \| P_X \otimes P_Y) = \mathbb{E}_{p(x,y)} \left[ \log \frac{p(x, y)}{p(x)p(y)} \right]. \tag{5}$$

The InfoMax principle chooses a representation $f(x)$ by maximizing the mutual information between the input $x$ and the representation $f(x)$. However, estimating MI, especially in high-dimensional spaces is challenging in nature. And one often optimizes a tractable lower bound of MI in practice (Poole et al., 2019).

## 3 AN INFORMATION-THEORETIC PERSPECTIVE ON DIFFUSION REPRESENTATION LEARNING

Despite some empirical attempts at Diffusion Representation Learning (DRL), its theoretical foundations remain largely uncharted. In this section, we analyze the DRL through the lens of Information Theory, establishing a connection between the DRL objective and mutual information.

### 3.1 THE ROLE OF EXTRA INFORMATION IN IMPROVING RECONSTRUCTION

Conditional diffusion models exhibit superior generation quality and lower denoising score matching loss compared to their unconditional counterparts, as observed by (Dhariwal & Nichol, 2021; Zhang et al., 2022). Figure 2 illustrates the denoising score matching loss for the label conditional task (**Label** curve) is lower than that for the unconditional task (**Vanilla** curve). This improvement is attributed to the additional information provided by class labels, which aids the diffusion model in effectively denoising noisy data. One might consider class labels $c$ as a special feature extracted from data: $c = E_\phi(\mathbf{x})$ where $E_\phi$ is a classifier that outputs class labels. This leads to speculation that more informative representations further enhance the denoising process and lower the denoising score matching loss conditioned on the representations. Thus intuitively one can jointly train the diffusion model conditioning on an additional feature extractor $E_\phi$ (Abstreiter et al., 2021; Hudson

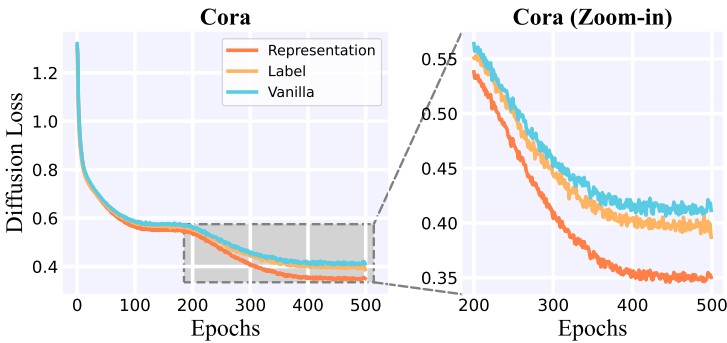

Figure 2: The comparison of denoising losses using different conditions on Cora datasets. **(Vanilla)** The denoising loss without condition information. **(Label)** Class label information obtained via linear embedding. **(Representation)** Learned representations obtained from `Graffe`.

et al., 2024), as the reconstruction denoising loss will guide the feature extractor to produce more informative representations. Formally, the learning objective for DRL is as follows:

$$\mathcal{L}_{\mathbf{x}_0,DSM,\phi} = \mathbb{E}_t \left\{ \lambda(t) \mathbb{E}_{\mathbf{x}_0} \mathbb{E}_{\mathbf{x}_t|\mathbf{x}_0} \left[ \|\mathbf{x}_\theta(\mathbf{x}_t, t, E_\phi(\mathbf{x}_0)) - \mathbf{x}_0\|^2 \right] \right\}. \tag{6}$$

In the next part of this section, we elucidate the intuition that more informative representations lead to lower denoising score matching loss from a theoretical standpoint. We eliminate the effects of limited network capacity or optimization errors, allowing us to investigate the influence of additional conditions on the denoising score matching loss under ideal conditions—specifically when the network capacity is adequate and optimization achieves its optimal state. The following theorem demonstrates that the denoising score matching objective has a positive lower bound, even when the network's capacity is sufficiently large.

**Theorem 1.** *The denoising score matching objective $\mathcal{L}_{\mathbf{x}_0,DSM}$ has a **strictly positive** lower bound, regardless of the network capacity and expressive power*

$$\begin{aligned} \min_{\mathbf{x}_\theta} \mathcal{L}_{\mathbf{x}_0,DSM} &= \min_{\mathbf{x}_\theta} \mathbb{E}_t \left\{ \lambda(t) \mathbb{E}_{\mathbf{x}_0} \mathbb{E}_{\mathbf{x}_t|\mathbf{x}_0} \left[ \|\mathbf{x}_\theta(\mathbf{x}_t, t) - \mathbf{x}_0\|^2 \right] \right\} \\ &= \mathbb{E}_t \left\{ \lambda(t) \mathbb{E}_{\mathbf{x}_t} \left[ \mathrm{Tr}(\mathrm{Cov}[\mathbf{x}_0|\mathbf{x}_t]) \right] \right\} > 0, \end{aligned} \tag{7}$$

*where $\mathrm{Tr}$ is the Trace of matrix and $\mathrm{Cov}$ is the covariance matrix. The conditioned denoising score matching objective objective $\mathcal{L}_{\mathbf{x}_0,DSM,\phi}$ has a **non-negative** lower bound, i.e.*

$$\min_{\mathbf{x}_\theta} \mathcal{L}_{\mathbf{x}_0,DSM,\phi} = \mathbb{E}_t \left\{ \lambda(t) \mathbb{E}_{\mathbf{x}_0,\mathbf{x}_t} \left[ \mathrm{Tr}(\mathrm{Cov}[\mathbf{x}_0|\mathbf{x}_t, E_\phi(\mathbf{x}_0)]) \right] \right\} \geq 0. \tag{8}$$

The proof is in Appendix A. Theorem 1 reveals an attractive property of the denoising score matching loss: its minimum value is determined by the uncertainty of the conditional distribution (the trace of the covariance matrix serves as a multidimensional generalization of variance). Additionally, Theorem 2 demonstrates that the supplementary information provided by the feature extractor $E_\phi$ reduces the lower bound of DSM by decreasing the uncertainty of the conditional distribution through more informative representations.

**Theorem 2.** *The conditioned denoising score matching objective $\mathcal{L}_{\mathbf{x}_0,DSM,\phi}$ has a smaller minimum compared with the vanilla objective:*

$$\min_{\mathbf{x}_\theta} \mathcal{L}_{\mathbf{x}_0,DSM,\phi} \leq \min_{\mathbf{x}_\theta} \mathcal{L}_{\mathbf{x}_0,DSM}. \tag{9}$$

The proof is in Appendix A. Theorem 2 offers a qualitative insight, indicating that informative representations diminish the uncertainty in the conditional distribution. Figure 2 shows the denoising score matching loss for the representation conditional task (**Representation** curve) is lower than both the unconditional task (**Vanilla** curve) and the label conditional task (**Label** curve). This suggests that the learned representation contains richer information than class labels alone.

### 3.2 DIFF-INFOMAX PRINCIPLE

Intuitively a poor representation dominated by noise provides little useful information, failing to assist the diffusion model in denoising. In contrast, a rich and informative representation enhances

the model's denoising capabilities. In this section, we will quantitatively analyze this from an information-theoretic perspective. Notably, the DRL objective is closely related to the conditional mutual information between $E_\phi(\mathbf{x}_0)$ and $\mathbf{x}_0$ given $\mathbf{x}_t$.

**Theorem 3.** *Suppose $\mathbf{x}_0 \in \mathbb{R}^d$, let $\mathcal{L}_{\mathbf{x}_0,DSM,\phi,t} = \mathbb{E}_{\mathbf{x}_0,\mathbf{x}_t}\left[\mathrm{Tr}(\mathrm{Cov}[\mathbf{x}_0|\mathbf{x}_t, E_\phi(\mathbf{x}_0)])\right]$ be the conditional denoising score matching loss at time $t$, and let $h(\mathbf{x}|\mathbf{y})$ be the conditional entropy of $\mathbf{x}$ given $\mathbf{y}$, then the negative logarithm of denoising score matching loss is a lower bound for the conditional mutual information between data and feature, which quantifies the shared information between $\mathbf{x}_0$ and $E_\phi(\mathbf{x}_0)$, given the knowledge of $\mathbf{x}_t$*

$$I(\mathbf{x}_0; E_\phi(\mathbf{x}_0)|\mathbf{x}_t) \geq -\log \mathcal{L}_{\mathbf{x}_0,DSM,\phi,t} + C, \quad \text{where } C = \log \frac{d}{2\pi e} + \frac{2}{d}h(\mathbf{x}_0|\mathbf{x}_t) \text{ is a constant.}$$
(10)

The proof is in Appendix A. Theorem 3 indicates that minimizing the diffusion reconstruction objective is equivalent to maximizing a lower bound of conditional mutual information between data and feature. Figure 3 illustrates the correlation between diffusion reconstruction loss and linear probing accuracy on downstream tasks. As the diffusion loss decreases, the lower bound of conditional mutual information increases, which in turn corresponds to higher linear probing accuracy. This supports our theory that a lower diffusion loss is associated with more informative representations, leading to improved performance in linear probing on downstream tasks.

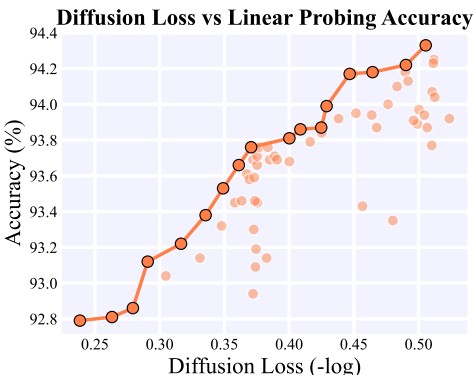

Figure 3: The correlation between the negative logarithm of diffusion loss (**x-axis**) and linear probing accuracy (**y-axis**) on the Photo dataset.

InfoMax principle (Linsker, 1988; Hjelm et al., 2018) proposes to choose a representation $f(\mathbf{x})$ by maximizing $I(\mathbf{x}; f(\mathbf{x}))$. Motivated by Theorem 3, we propose the Diff-InfoMax principle:

**Diff-InfoMax principle**   Choosing a representation $f(\mathbf{x})$ by maximizing $\int_0^t \lambda(t)I(\mathbf{x}; f(\mathbf{x})|\mathbf{x}_t)\mathrm{d}t$, where $\mathbf{x}_t = \alpha_t\mathbf{x} + \sigma_t\xi$ is a data corrupted by Gaussian Noise and $\lambda(t) \in \mathbb{R}$ is a weighting function.

The first key distinction between the Diff-InfoMax principle and the original InfoMax principle is that Diff-InfoMax optimizes the conditional mutual information $I(\mathbf{x}; f(\mathbf{x})|\mathbf{x}_t)$, which quantifies the shared information between $\mathbf{x}$ and $f(\mathbf{x})$, given the knowledge of $\mathbf{x}_t$. The second difference lies in Diff-InfoMax's use of a multi-level criterion, encouraging the representation to maximize information about $\mathbf{x}$ while excluding the information from $\mathbf{x}_t$. By accounting for different noise levels in $\mathbf{x}_t$, $I(\mathbf{x}; f(\mathbf{x})|\mathbf{x}_t)$ promotes the representation to capture varying levels of structural detail. Furthermore, we demonstrate that the original InfoMax principle is a special case of the proposed Diff-InfoMax principle.

**Remark 1.** *The original InfoMax principle can be viewed as a special case of the Diff-InfoMax principle when $\lambda(t) = \delta_T(t)$. Then $\int_0^T \delta_T(t)I(\mathbf{x}; f(\mathbf{x})|\mathbf{x}_t)\mathrm{d}t = I(\mathbf{x}; f(\mathbf{x})|\mathbf{x}_T) = I(\mathbf{x}; f(\mathbf{x}))$ because $\mathbf{x}_T$ is a pure Gaussian noise and independent with $\mathbf{x}$ and $f(\mathbf{x})$.*

Similar to MI, estimating conditional MI is particularly challenging in high-dimensional spaces. We address this by optimizing a tractable lower bound of conditional MI, specifically the DRL objective. We believe that the Diff-InfoMax principle opens up new avenues for integrating diffusion models with representation learning. Additionally, there are alternative methods for optimizing novel variational lower bounds of the conditional MI objective, which we reserve for future exploration.

### 3.3 EFFECTS ON FREQUENCY DOMAIN

**Frequency-aware Analysis**   Several works (Yang et al., 2023; Si et al., 2024; Dieleman, 2024) have noted that during the noising process, the high-frequency components of the data are corrupted first, followed by the low-frequency components. Conversely, in the generation process, low-frequency components are generated initially, with high-frequency components added later. Then the diffusion

model performs a role generating high-frequency components given noisy data which mainly consists of low-frequency data. From this frequency domain perspective, $I(\mathbf{x}; f(\mathbf{x})|\mathbf{x}_t)$ guides the feature extractor to focus on components with frequencies exceeding a certain threshold, with different time $t$ corresponding to different frequency thresholds.

**Graph Feature**  BWGNN (Tang et al., 2022) defines a metric *Energy Ratio* to assess the concentration of graph features in low frequencies. They observe that perturbing graph features with random noise results in a 'right-shift' of energy, indicating a reduced concentration in low frequencies and an increased concentration in high frequencies. This finding aligns with our analysis of the frequency domain. Consequently, DRL operates in the spectral space of graph features, excelling at capturing high-frequency information in these features."

## 4  THE **GRAFFE** APPROACH

As inspired by the above theoretical insights and to overcome the challenges mentioned in section 1, the **Graffe** framework follows the *explicit-encoder pattern* and couples a graph encoder $E_\phi$ with a conditional diffusion decoder $D_\theta$. Given an input graph $\mathcal{G} = (\mathbf{X}, \mathbf{A})$, the encoder achieves perception of both structural and feature information and extracts a compact representation $\mathbf{z} = E_\phi(\mathcal{G})$ for each node. Then, the decoder receives both noisy feature $\mathbf{x}_t$ and encoded representation $\mathbf{z}$ to reconstruct the original feature $\tilde{\mathbf{x}} = D_\theta(\mathbf{x}_t, t, \mathbf{z})$. The overall framework is demonstrated in fig. 1. We next introduce the **Graffe** in detail.

### 4.1  THE GRAPH ENCODER

The encoder module is the core part of our model. Since we are not concerned with generative capabilities, the encoder is the only parameterized module used in downstream tasks, and its capability directly impacts task performance. We consider two factors that guide the training lean toward representation learning: one is the **expressive capacity of the encoder**, which refers to whether it can fully perceive graph data to provide strong representations. The other is the **adequacy of encoder training**, which involves whether the optimization of the objective function can effectively coordinate the optimization of both the encoder and decoder.

For the first factor, we follow prior work Hou et al. (2022; 2023); Zhao et al. (2024) on the encoder selection, which adopted GAT (Velickovic et al., 2017) and GIN (Xu et al., 2018) for node and graph tasks, respectively, as both theoretical and empirical evidence demonstrate that they have strong expressive capabilities for graph tasks. This also ensures fair comparison in subsequent experimental analysis. Specifically, their message-passing mechanism can be expressed as:

$$h_v^{(k)} = \texttt{COMB}\left(h_v^{(k-1)}, \texttt{AGGR}\{h_u^{(k-1)} : u \in \mathcal{N}(v)\}\right), \quad 1 \leq k \leq L, \tag{11}$$

where $h_v^{(k)}$ denotes representation of node $v$ at the $k$-th layer, $\mathcal{N}(v)$ is the set of neighboring nodes connected to node $v$ and $L$ is the number of layers. $\texttt{AGGR}(\cdot)$ and $\texttt{COMB}(\cdot)$ are used for aggregating neighborhood information and combining ego- and neighbor-representations, respectively. For graph-level tasks, the $\texttt{READOUT}(\cdot)$ function aggregates node features from the final iteration to obtain the entire graph's representation.

It is worth noting that even given a powerful representation learner, there is a potential risk that the model training may tend to ignore the information in $\mathbf{z}$. This is because the input $\mathbf{x}$ to the encoder and the reconstruction target by the decoder are the same, which might lead the model to learn a *"shortcut"*. Consider an extreme case where the encoder performs an identity matrix mapping $E_\phi(\cdot) = \mathcal{I}(\cdot)$ on the input features, the optimization objective transforms to $\mathcal{L}_{\mathbf{x}_0, DSM} = \mathbb{E}_t\left\{\lambda(t)\mathbb{E}_{\mathbf{x}_0}\mathbb{E}_{\mathbf{x}_t|\mathbf{x}_0}\left[\|\mathbf{x}_\theta(\mathbf{x}_t, t, \mathbf{x}_0) - \mathbf{x}_0\|^2\right]\right\}$. In this scenario, the encoder obtains a poor capability to extract graph semantics, since the loss can easily approach zero. To this end, we randomly zero out partial node features before inputting them into the encoder.

Formally, let $\mathbf{X} \in \mathbb{R}^{n \times d}$ be a feature matrix. Define a masking vector $h_{[mask]}$ consisting of $n$ Bernoulli random variables with probability $m$, then the modified matrix $\mathbf{X}'$ can be expressed as:

$$h_{[mask]} \sim \text{Bernoulli}(1-m)^n, \quad \mathbf{X}' = \text{diag}(h_{[mask]})\mathbf{X}. \tag{12}$$

Using corrupted node features as input not only effectively prevents the model from learning shortcuts, but also reduces redundancy in attributed graphs. This approach essentially creates a more challenging self-supervision task for learning robust and meaningful representations.

## 4.2 THE DIFFUSION DECODER

**Reconstruction objective.** Unlike image features, graph data incorporates feature and structural information, prompting the question of which to prioritize for reconstruction. Previous work in graph SSL has explored both directions: for example, GraphMAE (Hou et al., 2022) focuses only on feature information, while another concurrent work, MaskGAE (Li et al., 2023), only targets topological attributes. It is worth noting that in many graph learning datasets, features are often one-hot embeddings, and topology is represented by adjacency matrices—both of which are highly sparse, thus making it difficult to make decisions based on the nature of data. We empirically tested reconstructing features, topology, and their combination. Results in table 3 demonstrate that feature reconstruction performs best, outperforming the hybrid approach, with topology-only reconstruction yielding the worst results. Therefore, we choose features $\mathbf{x}$ as the target for reconstruction.

**Customized instantiation of decoder.** In decoder design, we draw on the experience of using the U-Net architecture from the visual domain as a backbone model for diffusion training. The U-Net architecture Ronneberger et al. (2015) provides representations of different granularities through up- and down-sampling Si et al. (2024). Additionally, it aligns well with the strict dimensional requirements of diffusion models. Specifically, when handling graph-level tasks, we propose Graph-UNet, which adopts GNN layers to replace the convolutional layers in the vanilla U-Net. In this context, each graph in a mini-batch can be likened to an image in a visual diffusion model; by uniformly sampling time step $t \sim \text{Uniform}(0, T)$ within a mini-batch, we ensure that the level of feature noise within each graph remains consistent.

However, for node-level tasks, if we instantiate the decoder with GNNs, it becomes problematic to use different time steps for different nodes, as this would lead to message passing propagating node information at varying noise levels. Therefore, to enable the model to clearly perceive distinct noise levels and conduct training in a principled manner, we replace the GNN layers with the MLP network. Please refer to appendix B for more details of Graph-Unet.

## 5 EXPERIMENTS

### 5.1 EXPERIMENTAL SETUP

**Datasets.** Our experiments primarily involve node-level and graph-level datasets. For node classification tasks, we select 6 datasets drawn from various domains for evaluation. These include three citation networks: `Cora`, `CiteSeer`, and `PubMed` Sen et al. (2008); two co-purchase graphs: `Photo` and `Computer` Shchur et al. (2018); and a large dataset from the Open Graph Benchmark: `arXiv` Hu et al. (2020a). The above evaluation datasets represent real-world networks and graphs from diverse fields. For graph classification tasks, we select 5 datasets for training and testing: `IMDB-B`, `IMDB-M`, `PROTEINS`, `COLLAB`, and `MUTAG` Yanardag & Vishwanathan (2015). Each dataset comprises a collection of graphs, with each graph assigned a label. In graph classification tasks, the node degrees are used as attributes for all datasets. These features are further processed using one-hot encoding as input to the model.

**Evaluation protocols.** We follow the experimental settings from (Hassani & Khasahmadi, 2020; Velickovic et al., 2019). First, we train a GNN encoder and a decoder using the proposed `Graffe` in an unsupervised manner. Then, we freeze the encoder parameters to infer the node representations. We train a linear classifier to evaluate the representation quality and report the average accuracy on test nodes over 20 random initializations. For node classification tasks, we use the public data splits of `Cora`, `Citeseer`, and `PubMed` as specified in (Hassani & Khasahmadi, 2020; Thakoor et al., 2021; Velickovic et al., 2019) and adopt GAT (Velickovic et al., 2017) as the graph encoder. For graph classification tasks, we follow the experimental setup by Hou et al. (2022) and adopt the GIN (Xu et al., 2018) as the graph encoder. We feed the graph-level representations into the downstream LIBSVM classifier Chang & Lin (2001) to predict labels. The average 10-fold cross-validation accuracy and standard deviation after 5 runs.

Table 1: Empirical performance of self-supervised representation learning for node classification in terms of accuracy (%, ↑). We highlight the best- and the second-best performing results in **boldface** and underlined, respectively.

| | Dataset | Cora | CiteSeer | PubMed | Ogbn-arxiv | Computer | Photo |
|---|---|---|---|---|---|---|---|
| Supervised | GCN | 81.5±0.5 | 70.3±0.7 | 79.0±0.4 | 71.7±0.3 | 86.5±0.5 | 92.4±0.2 |
| | GAT | 83.0±0.7 | 72.5±0.7 | 79.0±0.3 | 72.1±0.1 | 86.9±0.3 | 92.6±0.4 |
| Self-supervised | GAE | 71.5±0.4 | 65.8±0.4 | 72.1±0.5 | 63.6±0.5 | 85.1 ± 0.4 | 91.0±0.2 |
| | GPT-GNN | 80.1±1.0 | 68.4±1.6 | 76.3±0.8 | - | - | - |
| | GATE | 83.2±0.6 | 71.8±0.8 | 80.9±0.3 | - | - | - |
| | DGI | 82.3±0.6 | 71.8±0.7 | 76.8±0.6 | 70.3±0.2 | 84.0±0.5 | 91.6±0.2 |
| | MVGRL | 83.5±0.4 | 73.3±0.5 | 80.1±0.7 | - | 87.5±0.1 | 91.7±0.1 |
| | GRACE | 81.9±0.4 | 71.2±0.5 | 80.6±0.4 | 71.5±0.1 | 86.3±0.3 | 92.2±0.2 |
| | BGRL | 82.7±0.6 | 71.1±0.8 | 79.6±0.5 | 71.6±0.1 | 89.7±0.3 | 92.9±0.3 |
| | InfoGCL | 83.5±0.3 | 73.5 ±0.4 | 79.1±0.2 | - | - | - |
| | CCA-SSG | 84.0±0.4 | 73.1±0.3 | 81.0±0.4 | 71.2±0.2 | 88.7±0.3 | 93.1±0.1 |
| | GraphMAE | 84.2±0.4 | 73.4±0.4 | 81.1±0.4 | 71.8±0.2 | 88.6±0.2 | 93.6 ± 0.2 |
| | GraphMAE2 | 84.1±0.6 | 73.1±0.4 | 80.9±0.5 | 71.8±0.0 | 89.2±0.4 | 93.3 ± 0.2 |
| | MaskGAE$_{edge}$ | 83.8±0.3 | 72.9±0.2 | 82.7±0.3 | 71.0±0.3 | 89.4±0.1 | 93.3 ± 0.0 |
| | MaskGAE$_{path}$ | 84.3±0.3 | 73.8±0.8 | 83.6±0.5 | 71.2±0.3 | 89.5±0.1 | 93.3 ± 0.1 |
| | DDM | 83.4±0.2 | 72.5±0.3 | 79.6±0.8 | 71.3±0.2 | 89.9±0.2 | 93.8±0.2 |
| | Bandana | 84.5±0.3 | 73.6±0.2 | **83.7±0.5** | 71.1±0.2 | 89.6±0.1 | 93.4 ± 0.1 |
| | **Graffe** | **84.8±0.4** | **74.3±0.4** | 81.0±0.6 | **72.1±0.2** | **91.3±0.2** | **94.2±0.1** |

**Implementation details.** In our study, we employ either Adam (Kingma, 2014) or AdamW (Loshchilov, 2017) as the optimizer, complemented by a cosine annealing scheduler (Loshchilov & Hutter, 2016) to enhance model convergence across different datasets. Moreover, we configure the learning rate for the encoder to be twice that of the decoder, a strategy that has demonstrated empirical effectiveness in promoting training stability. In terms of the noise schedule, we explore several candidate approaches, including sigmoid, linear, and inverted schedules, ultimately selecting the most appropriate method based on their performance for each dataset. Detailed hyper-parameter configurations are provided in the appendix C.

## 5.2 NODE CLASSIFICATION

For comprehensive comparison, we select the following three groups of SSL methods as primary baselines in our experiments. ① Auto-encoding methods: GAE (Kipf & Welling, 2016), GATE (Salehi & Davulcu, 2019), GraphMAE(Hou et al., 2022), GraphMAE2(Hou et al., 2023), MaskGAE(Li et al., 2023), Bandana(Zhao et al., 2024) ② Contrastive methods: GRACE (Zhu et al., 2021), CCA-SSG (Zhang et al., 2021), InfoGCL (Xu et al., 2021), DGI(Velickovic et al., 2019), MVGRL (Hassani & Khasahmadi, 2020), BGRL (Thakoor et al., 2021), GCC (Qiu et al., 2020) ③ Others: GPT-GNN (Hu et al., 2020b), DDM (Yang et al., 2024). The performance of 6 linear probing node classification tasks is summarized in table 1. The results not reported are due to unavailable code or out-of-memory. Generally, it can be found from the table that our **Graffe** shows strong empirical performance across all datasets, delivering five out of six state-of-the-art results. The outstanding results validate the superiority of our proposed model.

We make other observations as follows: *(i)* Note that previous work has already achieved pretty high performance. For example, the current state-of-the-art DDM only obtains a 0.24% absolute improvement over the second-best baseline, Bandana, in terms of average accuracy on the `Computer` dataset. Our work pushes that boundary with absolute improvement up to 1.46% over DDM. *(ii)* Our method surpasses the supervised training baseline on almost all tasks. For instance, in the `Computer` dataset, the GAT baseline achieves an accuracy of 86.9 under fully supervised training; however, **Graffe** improves upon this by 4.4 percentage points. Interestingly, this further corroborates our theoretical findings presented in section 3.1 and illustrated in fig. 2. It demonstrates that our proposed model is able to obtain meaningful and high-quality embeddings.

## 5.3 GRAPH CLASSIFICATION

For graph classification tasks, we further include the graph kernel methods (Shervashidze et al., 2011; Yanardag & Vishwanathan, 2015) and graph2vec (Narayanan et al., 2017) following Hou et al. (2022).

Table 2: Experiment results in self-supervised representation learning for graph classification. We report accuracy (%) for all datasets. We highlight the best- and the second-best performing results in **boldface** and underlined, respectively.

| | Dataset | IMDB-B | IMDB-M | PROTEINS | COLLAB | MUTAG |
|---|---|---|---|---|---|---|
| Supervised | GIN | 75.1±5.1 | 52.3±2.8 | 76.2±2.8 | 80.2±1.9 | 89.4±5.6 |
| | DiffPool | 72.6±3.9 | - | 75.1±3.5 | 78.9±2.3 | 85.0±10.3 |
| Graph Kernels | WL | 72.30±3.44 | 46.95±0.46 | 72.92±0.56 | - | 80.72±3.00 |
| | DGK | 66.96±0.56 | 44.55±0.52 | 73.30±0.82 | - | 87.44±2.72 |
| Self-supervised | graph2vec | 71.10±0.54 | 50.44±0.87 | 73.30±2.05 | - | 83.15±9.25 |
| | Infograph | 73.03±0.87 | 49.69±0.53 | 74.44±0.31 | 70.65±1.13 | 89.01±1.13 |
| | GraphCL | 71.14±0.44 | 48.58±0.67 | 74.39±0.45 | 71.36±1.15 | 86.80±1.34 |
| | JOAO | 70.21±3.08 | 49.20±0.77 | 74.55±0.41 | 69.50±0.36 | 87.35±1.02 |
| | GCC | 72.0 | 49.4 | - | 78.9 | - |
| | MVGRL | 74.20±0.70 | 51.20±0.50 | - | - | 89.70±1.10 |
| | InfoGCL | 75.10±0.90 | 51.40±0.80 | - | 80.00±1.30 | 91.20±1.30 |
| | GraphMAE | 75.52±0.66 | 51.63±0.52 | **75.30±0.39** | 80.32±0.46 | 88.19±1.26 |
| | DDM | 74.05±0.17 | 52.02±0.29 | 71.61±0.56 | 80.70±0.18 | 90.15±0.46 |
| | **Graffe** | **76.20±0.23** | **52.4±0.37** | 74.36±0.12 | **81.28±0.15** | **91.46±0.26** |

The performance of **Graffe** on 5 datasets is summarized in table 2. It can be observed that our method demonstrates performant results on different tasks, achieving state-of-the-art results on 4 out of 5 datasets. This further indicates that **Graffe**, as a new class of generative SSL, holds significant potential in representation learning. Furthermore, similar to observations in node classification, our method also outperforms fully supervised counterparts.

## 5.4 Ablation Study

**Effect of different components** To demonstrate the necessity of each module in our model, we conduct ablation study to validate the different components of **Graffe**. Specifically, we consider three aspects for ablation: reconstruction objectives, masking strategies, and decoder selection. We select Cora, Computer, and Photo for node-level tasks, and IMDB-B, COLLAB, and MUTAG for graph-level tasks. The experimental results are presented in table 3. Our observations are as follows: *(i)* The performance of reconstructing only feature (i.e., the Graffe model) surpasses that of the mixed reconstruction, with the worst performance occurring when reconstructing only topology.

Table 3: Ablation of different components.

| Node-level | Cora | Computer | Photo |
|---|---|---|---|
| **A** Recons. | 77.6 | 86.2 | 91.7 |
| **A** + **X** Recons. | 80.1 | 87.4 | 92.2 |
| w/o Mask | 82.5 | 88.5 | 92.5 |
| w. GAT decoder | 83.2 | 89.8 | 92.9 |
| **Graffe** | **84.8** | **91.3** | **94.2** |

| Graph-level | IMDB-B | COLLAB | MUTAG |
|---|---|---|---|
| **A** Recons. | 70.2 | 71.5 | 83.6 |
| **A** + **X** Recons. | 71.6 | 77.6 | 86.8 |
| w/o Mask | 75.8 | 81.2 | 91.5 |
| w. MLP decoder | 74.5 | 79.9 | 88.5 |
| **Graffe** | **76.2** | **81.3** | **91.5** |

This suggests that explicitly reconstructing structural information leads to performance degradation. *(ii)* The masking strategy is particularly critical for node-level tasks, as its removal results in significant performance drops, while the impact is less noticeable for graph-level tasks. *(iii)* The choice of decoder layers is critical for different task types. For node-level tasks, using an MLP layer yields better results compared to a GAT layer, while the opposite is true for graph-level tasks. This aligns with our intuitive analysis in section 4.2, indicating that the propagation of noise is detrimental to diffusion representation learning.

**Effect of mask ratio** Since mask strategy is a crucial component of our framework, it is necessary to evaluate how to choose a proper $m$. We conduct an empirical analysis on Cora, Computer and MUTAG dataset and consider a candidate list covering the value ranges of $m$: [0, 0.1, 0.3, 0.5, 0.7, 0.9]. As shown in fig. 4, the optimal masking choice varies across different datasets. For the Cora and Computer datasets, the best performance is achieved when $m = 0.7$, whereas on the MUTAG dataset, the best results are obtained without applying any masking.

Figure 4: The effect of mask ratio $m$.

Moreover, a higher mask ratio even leads to performance decline on graph-level tasks. This suggests that the selection of the mask ratio should be tuned according to the specific task requirements, as there is no one-size-fits-all solution.

# 6 RELATED WORK

## 6.1 SELF-SUPERVISED LEARNING ON GRAPHS

**Contrastive methods** Being popular in SSL, contrastive methods aim to learn discriminative representations by contrasting positive and negative samples. The key to obtain distinguishable representations lies in the way of constructing contrastive pairs. DGI (Velickovic et al., 2019) and InfoGraph (Sun et al., 2019), based on MI maximization, corrupt graph feature and topology to construct negative samples. To avoid the underlying risk of semantic damage, GRACE (Zhu et al., 2020), GCA (Zhu et al., 2021), and GraphCL (You et al., 2020) use other graphs within the same batch as negatives. Other works, i.e., BGRL (Thakoor et al., 2021) and CCA-SSA (Zhang et al., 2021), propose to achieve contrastive learning free of negatives yet demanding strong regularization or feature decorrelation. A line of works borrow from data augmentation in the field of computer vision (CV) to construct constrastive pairs, including feature-oriented ((Thakoor et al., 2021; You et al., 2020; Zhu et al., 2020), shuffling (Velickovic et al., 2019)), perturbation (Hu et al., 2020b; You et al., 2020)), and graph-theory-based (random walk (Hassani & Khasahmadi, 2020; Qiu et al., 2020).

**Generative methods** Generative self-supervised methods aim to learn informative representations using learning signals from the data itself, usually by maximizing the marginal log-likelihood of the data. GPT-GNN Hu et al. (2020b), following the auto-regressive paradigm, iteratively generates graph features and topology, which is unnatural as most graph data has no inherent order. GAE and VGAE Kipf & Welling (2016) learn to reconstruct the adjacency matrix by using the representation learned from GCN, while other graph autoencoders Salehi & Davulcu (2019); Hou et al. (2022) further combine it with feature reconstruction with tailored strategies. However, these generative methods are usually not principled in terms of probabilistic generative models and often prove to be inferior to the contrastive ones.

## 6.2 DIFFUSION MODELS FOR REPRESENTATION LEARNING

The very first attempt has combined auto-encoders with diffusion models—e.g., DiffAE (Preechakul et al., 2022), a non-probabilistic auto-encoder model that produces semantically meaningful latent. InfoDiffusion (Wang et al., 2023), as the first principled probabilistic generative model for representation learning, augments DiffAE with an auxiliary-variable model family and mutual information maximization. Similarly, Zhang et al. (2022) uses a pre-trained diffusion decoder and designs a re-weighting scheme to fill in the posterior mean gap. Targeting image classification tasks, Wei et al. (2023); Gao et al. (2023); Hudson et al. (2024) combine latent diffusion with the self-supervised learning objective to get meaningful representations. The decoder-only models (Xiang et al., 2023; Chen et al., 2024), directly use the representations from intermediate layers without auxiliary encoders. However, the use of expressive diffusion models for graph representation learning remains under-explored. DDM (Yang et al., 2024) takes an initial step, but the proposed diffusion process is not mathematically rigorous and principled.

# 7 CONCLUSION

In this paper, we introduce `Graffe`, a self-supervised diffusion representation learning (DRL) framework designed for graphs, achieving state-of-the-art performance on self-supervised graph representation learning tasks. We establish the theoretical foundations of DRL and prove that the denoising objective is a lower bound for the conditional mutual information between data and its representations. We propose the Diff-InfoMax principle, an extension of the standard InfoMax principle, and demonstrate that DRL implicitly follows it. Based on these theoretical insights and customized design for graph data, `Graffe` excels in node and graph classification tasks. We provide discussion about limitations and future work in appendix C.

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

# A  PROOFS OF THEOREMS

## A.1  PROOF OF THEOREM 1

**Theorem 1.** *The denoising score matching objective $\mathcal{L}_{\mathbf{x}_0,DSM}$ has a **strictly positive** lower bound, regardless of the network capacity and expressive power*

$$\min_{\mathbf{x}_\theta} \mathcal{L}_{\mathbf{x}_0,DSM} = \min_{\mathbf{x}_\theta} \mathbb{E}_t \left\{ \lambda(t) \mathbb{E}_{\mathbf{x}_0} \mathbb{E}_{\mathbf{x}_t|\mathbf{x}_0} \left[ \|\mathbf{x}_\theta(\mathbf{x}_t,t) - \mathbf{x}_0\|^2 \right] \right\}$$

$$= \mathbb{E}_t \left\{ \lambda(t) \mathbb{E}_{\mathbf{x}_t} \left[ \text{Tr}(\text{Cov}[\mathbf{x}_0|\mathbf{x}_t]) \right] \right\} > 0. \tag{13}$$

*The conditioned denoising score matching objective objective $\mathcal{L}_{\mathbf{x}_0,DSM,\phi}$ has a **non-negative** lower bound, i.e.*

$$\min_{\mathbf{x}_\theta} \mathcal{L}_{\mathbf{x}_0,DSM,\phi} = \mathbb{E}_t \left\{ \lambda(t) \mathbb{E}_{\mathbf{x}_0,\mathbf{x}_t} \left[ \text{Tr}(\text{Cov}[\mathbf{x}_0|\mathbf{x}_t, E_\phi(\mathbf{x}_0)]) \right] \right\} \geq 0. \tag{14}$$

*Proof.*

$$\operatorname*{argmin}_{\mathbf{x}_\theta} \mathcal{L}_{\mathbf{x}_0,DSM}$$

$$= \operatorname*{argmin}_{\mathbf{x}_\theta} \mathbb{E}_t \left\{ \lambda(t) \mathbb{E}_{\mathbf{x}_0} \mathbb{E}_{\mathbf{x}_t|\mathbf{x}_0} \left[ \|\mathbf{x}_\theta(\mathbf{x}_t,t) - \mathbf{x}_0\|^2 \right] \right\}$$

$$= \operatorname*{argmin}_{\mathbf{x}_\theta} \mathbb{E}_t \left\{ \lambda(t) \mathbb{E}_{\mathbf{x}_0,\mathbf{x}_t} \left[ \|\mathbf{x}_\theta(\mathbf{x}_t,t) - \mathbb{E}[\mathbf{x}_0|\mathbf{x}_t] + \mathbb{E}[\mathbf{x}_0|\mathbf{x}_t] - \mathbf{x}_0\|^2 \right] \right\}$$

$$= \operatorname*{argmin}_{\mathbf{x}_\theta} \mathbb{E}_t \left\{ \lambda(t) \mathbb{E}_{\mathbf{x}_0,\mathbf{x}_t} \left[ \|\mathbf{x}_\theta(\mathbf{x}_t,t) - \mathbb{E}[\mathbf{x}_0|\mathbf{x}_t]\|^2 + 2\langle \mathbf{x}_\theta(\mathbf{x}_t,t) - \mathbb{E}[\mathbf{x}_0|\mathbf{x}_t], \mathbb{E}[\mathbf{x}_0|\mathbf{x}_t] - \mathbf{x}_0 \rangle \right] \right.$$

$$\left. + \lambda(t) \mathbb{E}_{\mathbf{x}_0,\mathbf{x}_t} \left[ \|\mathbb{E}[\mathbf{x}_0|\mathbf{x}_t] - \mathbf{x}_0\|^2 \right] \right\}$$

$$= \operatorname*{argmin}_{\mathbf{x}_\theta} \mathbb{E}_t \left\{ \lambda(t) \mathbb{E}_{\mathbf{x}_0,\mathbf{x}_t} \left[ \|\mathbf{x}_\theta(\mathbf{x}_t,t) - \mathbb{E}[\mathbf{x}_0|\mathbf{x}_t]\|^2 + 2\langle \mathbf{x}_\theta(\mathbf{x}_t,t) - \mathbb{E}[\mathbf{x}_0|\mathbf{x}_t], \mathbb{E}[\mathbf{x}_0|\mathbf{x}_t] - \mathbf{x}_0 \rangle \right] \right\}. \tag{15}$$

Note that

$$\mathbb{E}_{\mathbf{x}_0,\mathbf{x}_t} \left[ \langle \mathbf{x}_\theta(\mathbf{x}_t,t) - \mathbb{E}[\mathbf{x}_0|\mathbf{x}_t], \mathbb{E}[\mathbf{x}_0|\mathbf{x}_t] - \mathbf{x}_0 \rangle \right]$$

$$= \mathbb{E}_{\mathbf{x}_t} \mathbb{E}_{\mathbf{x}_0|\mathbf{x}_t} \left[ \langle \mathbf{x}_\theta(\mathbf{x}_t,t) - \mathbb{E}[\mathbf{x}_0|\mathbf{x}_t], \mathbb{E}[\mathbf{x}_0|\mathbf{x}_t] - \mathbf{x}_0 \rangle \right] \tag{16}$$

$$= \mathbb{E}_{\mathbf{x}_t} \left[ \langle \mathbf{x}_\theta(\mathbf{x}_t,t) - \mathbb{E}[\mathbf{x}_0|\mathbf{x}_t], \mathbb{E}_{\mathbf{x}_0|\mathbf{x}_t} [\mathbb{E}[\mathbf{x}_0|\mathbf{x}_t] - \mathbf{x}_0] \rangle \right].$$

Due to the property of conditional expectation, we have that

$$\mathbb{E}_{\mathbf{x}_0|\mathbf{x}_t} [\mathbb{E}[\mathbf{x}_0|\mathbf{x}_t] - \mathbf{x}_0] = \mathbb{E}[\mathbf{x}_0|\mathbf{x}_t] - \mathbb{E}[\mathbf{x}_0|\mathbf{x}_t] = 0. \tag{17}$$

Thus we have

$$\mathbb{E}_{\mathbf{x}_0,\mathbf{x}_t} \left[ \langle \mathbf{x}_\theta(\mathbf{x}_t,t) - \mathbb{E}[\mathbf{x}_0|\mathbf{x}_t], \mathbb{E}[\mathbf{x}_0|\mathbf{x}_t] - \mathbf{x}_0 \rangle \right] = 0. \tag{18}$$

Thus

$$\operatorname*{argmin}_{\mathbf{x}_\theta} \mathcal{L}_{\mathbf{x}_0,DSM}$$

$$= \operatorname*{argmin}_{\mathbf{x}_\theta} \mathbb{E}_t \left\{ \lambda(t) \mathbb{E}_{\mathbf{x}_0,\mathbf{x}_t} \left[ \|\mathbf{x}_\theta(\mathbf{x}_t,t) - \mathbb{E}[\mathbf{x}_0|\mathbf{x}_t]\|^2 + 2\langle \mathbf{x}_\theta(\mathbf{x}_t,t) - \mathbb{E}[\mathbf{x}_0|\mathbf{x}_t], \mathbb{E}[\mathbf{x}_0|\mathbf{x}_t] - \mathbf{x}_0 \rangle \right] \right\}$$

$$= \operatorname*{argmin}_{\mathbf{x}_\theta} \mathbb{E}_t \left\{ \lambda(t) \mathbb{E}_{\mathbf{x}_0,\mathbf{x}_t} \left[ \|\mathbf{x}_\theta(\mathbf{x}_t,t) - \mathbb{E}[\mathbf{x}_0|\mathbf{x}_t]\|^2 \right] \right\}$$

$$= \mathbb{E}[\mathbf{x}_0|\mathbf{x}_t]. \tag{19}$$

Substitute the minimizer of $\mathcal{L}_{\mathbf{x}_0,DSM}$ into it, we get the minimum of $\mathcal{L}_{\mathbf{x}_0,DSM}$

$$\min_{\mathbf{x}_\theta} \mathcal{L}_{\mathbf{x}_0,DSM}$$

$$= \min_{\mathbf{x}_\theta} \mathbb{E}_t \left\{ \lambda(t) \mathbb{E}_{\mathbf{x}_0} \mathbb{E}_{\mathbf{x}_t|\mathbf{x}_0} \left[ \|\mathbf{x}_\theta(\mathbf{x}_t,t) - \mathbf{x}_0\|^2 \right] \right\}$$

$$= \mathbb{E}_t \left\{ \lambda(t) \mathbb{E}_{\mathbf{x}_0} \mathbb{E}_{\mathbf{x}_t|\mathbf{x}_0} \left[ \|\mathbb{E}[\mathbf{x}_0|\mathbf{x}_t] - \mathbf{x}_0\|^2 \right] \right\}$$

$$= \mathbb{E}_t \left\{ \lambda(t) \mathbb{E}_{\mathbf{x}_t} \mathbb{E}_{\mathbf{x}_0|\mathbf{x}_t} \left[ (\mathbb{E}[\mathbf{x}_0|\mathbf{x}_t] - \mathbf{x}_0)^T (\mathbb{E}[\mathbf{x}_0|\mathbf{x}_t] - \mathbf{x}_0) \right] \right\} \tag{20}$$

$$= \mathbb{E}_t \left\{ \lambda(t) \mathbb{E}_{\mathbf{x}_t} \mathbb{E}_{\mathbf{x}_0|\mathbf{x}_t} \left[ \text{Tr}((\mathbb{E}[\mathbf{x}_0|\mathbf{x}_t] - \mathbf{x}_0)^T (\mathbb{E}[\mathbf{x}_0|\mathbf{x}_t] - \mathbf{x}_0)) \right] \right\}$$

$$= \mathbb{E}_t \left\{ \lambda(t) \mathbb{E}_{\mathbf{x}_t} \mathbb{E}_{\mathbf{x}_0|\mathbf{x}_t} \left[ \text{Tr}((\mathbb{E}[\mathbf{x}_0|\mathbf{x}_t] - \mathbf{x}_0)(\mathbb{E}[\mathbf{x}_0|\mathbf{x}_t] - \mathbf{x}_0)^T) \right] \right\}$$

$$= \mathbb{E}_t \left\{ \lambda(t) \mathbb{E}_{\mathbf{x}_t} \left[ \text{Tr}(\mathbb{E}_{\mathbf{x}_0|\mathbf{x}_t} [(\mathbb{E}[\mathbf{x}_0|\mathbf{x}_t] - \mathbf{x}_0)(\mathbb{E}[\mathbf{x}_0|\mathbf{x}_t] - \mathbf{x}_0)^T]) \right] \right\}$$

$$= \mathbb{E}_t \left\{ \lambda(t) \mathbb{E}_{\mathbf{x}_t} \left[ \text{Tr}(\text{Cov}[\mathbf{x}_0|\mathbf{x}_t]) \right] \right\} > 0.$$

The minimum is strictly positive for non-degenerated distributions $\mathbf{x}_0|\mathbf{x}_t$.

The proof of conditioned denoising score matching objective is similar.

$$
\begin{aligned}
&\operatorname*{argmin}_{\mathbf{x}_\theta} \mathcal{L}_{\mathbf{x}_0, DSM, \phi} \\
&= \operatorname*{argmin}_{\mathbf{x}_\theta} \mathbb{E}_t \left\{ \lambda(t) \mathbb{E}_{\mathbf{x}_0} \mathbb{E}_{\mathbf{x}_t|\mathbf{x}_0} \left[ \|\mathbf{x}_\theta(\mathbf{x}_t, t, E_\phi(\mathbf{x}_0)) - \mathbf{x}_0\|^2 \right] \right\} \\
&= \operatorname*{argmin}_{\mathbf{x}_\theta} \mathbb{E}_t \left\{ \lambda(t) \mathbb{E}_{\mathbf{x}_0, \mathbf{x}_t} \left[ \|\mathbf{x}_\theta(\mathbf{x}_t, t, E_\phi(\mathbf{x}_0)) - \mathbb{E}[\mathbf{x}_0|\mathbf{x}_t, E_\phi(\mathbf{x}_0)] + \mathbb{E}[\mathbf{x}_0|\mathbf{x}_t, E_\phi(\mathbf{x}_0)] - \mathbf{x}_0\|^2 \right] \right\} \\
&= \operatorname*{argmin}_{\mathbf{x}_\theta} \mathbb{E}_t \Big\{ \lambda(t) \mathbb{E}_{\mathbf{x}_0, \mathbf{x}_t} \left[ \|\mathbf{x}_\theta(\mathbf{x}_t, t, E_\phi(\mathbf{x}_0)) - \mathbb{E}[\mathbf{x}_0|\mathbf{x}_t, E_\phi(\mathbf{x}_0)]\|^2 \right] + \\
&\quad + 2\lambda(t) \mathbb{E}_{\mathbf{x}_0, \mathbf{x}_t} \left[ \langle \mathbf{x}_\theta(\mathbf{x}_t, t, E_\phi(\mathbf{x}_0)) - \mathbb{E}[\mathbf{x}_0|\mathbf{x}_t, E_\phi(\mathbf{x}_0)], \mathbb{E}[\mathbf{x}_0|\mathbf{x}_t, E_\phi(\mathbf{x}_0)] - \mathbf{x}_0 \rangle \right] \\
&\quad + \lambda(t) \mathbb{E}_{\mathbf{x}_0, \mathbf{x}_t} \left[ \|\mathbb{E}[\mathbf{x}_0|\mathbf{x}_t, E_\phi(\mathbf{x}_0)] - \mathbf{x}_0\|^2 \right] \Big\} \\
&= \operatorname*{argmin}_{\mathbf{x}_\theta} \mathbb{E}_t \Big\{ \lambda(t) \mathbb{E}_{\mathbf{x}_0, \mathbf{x}_t} \left[ \|\mathbf{x}_\theta(\mathbf{x}_t, t, E_\phi(\mathbf{x}_0)) - \mathbb{E}[\mathbf{x}_0|\mathbf{x}_t, E_\phi(\mathbf{x}_0)]\|^2 \right] \\
&\quad + 2\lambda(t) \mathbb{E}_{\mathbf{x}_0, \mathbf{x}_t} \left[ \langle \mathbf{x}_\theta(\mathbf{x}_t, t, E_\phi(\mathbf{x}_0)) - \mathbb{E}[\mathbf{x}_0|\mathbf{x}_t, E_\phi(\mathbf{x}_0)], \mathbb{E}[\mathbf{x}_0|\mathbf{x}_t, E_\phi(\mathbf{x}_0)] - \mathbf{x}_0 \rangle \right] \Big\}.
\end{aligned}
\tag{21}
$$

Note that

$$
\begin{aligned}
&\mathbb{E}_{\mathbf{x}_0, \mathbf{x}_t} \left[ \langle \mathbf{x}_\theta(\mathbf{x}_t, t, E_\phi(\mathbf{x}_0)) - \mathbb{E}[\mathbf{x}_0|\mathbf{x}_t, E_\phi(\mathbf{x}_0)], \mathbb{E}[\mathbf{x}_0|\mathbf{x}_t, E_\phi(\mathbf{x}_0)] - \mathbf{x}_0 \rangle \right] \\
&= \mathbb{E}_{\mathbf{x}_0, \mathbf{x}_t, E_\phi(\mathbf{x}_0)} \left[ \langle \mathbf{x}_\theta(\mathbf{x}_t, t, E_\phi(\mathbf{x}_0)) - \mathbb{E}[\mathbf{x}_0|\mathbf{x}_t, E_\phi(\mathbf{x}_0)], \mathbb{E}[\mathbf{x}_0|\mathbf{x}_t, E_\phi(\mathbf{x}_0)] - \mathbf{x}_0 \rangle \right] \\
&= \mathbb{E}_{\mathbf{x}_t, E_\phi(\mathbf{x}_0)} \mathbb{E}_{\mathbf{x}_0|\mathbf{x}_t, E_\phi(\mathbf{x}_0)} \left[ \langle \mathbf{x}_\theta(\mathbf{x}_t, t, E_\phi(\mathbf{x}_0)) - \mathbb{E}[\mathbf{x}_0|\mathbf{x}_t, E_\phi(\mathbf{x}_0)], \mathbb{E}[\mathbf{x}_0|\mathbf{x}_t, E_\phi(\mathbf{x}_0)] - \mathbf{x}_0 \rangle \right] \\
&= \mathbb{E}_{\mathbf{x}_t, E_\phi(\mathbf{x}_0)} \left[ \langle \mathbf{x}_\theta(\mathbf{x}_t, t, E_\phi(\mathbf{x}_0)) - \mathbb{E}[\mathbf{x}_0|\mathbf{x}_t, E_\phi(\mathbf{x}_0)], \mathbb{E}_{\mathbf{x}_0|\mathbf{x}_t, E_\phi(\mathbf{x}_0)} [\mathbb{E}[\mathbf{x}_0|\mathbf{x}_t, E_\phi(\mathbf{x}_0)] - \mathbf{x}_0] \rangle \right].
\end{aligned}
\tag{22}
$$

Due to the property of conditional expectation, we have that

$$
\mathbb{E}_{\mathbf{x}_0|\mathbf{x}_t, E_\phi(\mathbf{x}_0)} \left[ \mathbb{E}[\mathbf{x}_0|\mathbf{x}_t, E_\phi(\mathbf{x}_0)] - \mathbf{x}_0 \right] = \mathbb{E}[\mathbf{x}_0|\mathbf{x}_t, E_\phi(\mathbf{x}_0)] - \mathbb{E}[\mathbf{x}_0|\mathbf{x}_t, E_\phi(\mathbf{x}_0)] = 0.
\tag{23}
$$

Thus we have

$$
\mathbb{E}_{\mathbf{x}_0, \mathbf{x}_t} \left[ \langle \mathbf{x}_\theta(\mathbf{x}_t, t, E_\phi(\mathbf{x}_0)) - \mathbb{E}[\mathbf{x}_0|\mathbf{x}_t, E_\phi(\mathbf{x}_0)], \mathbb{E}[\mathbf{x}_0|\mathbf{x}_t, E_\phi(\mathbf{x}_0)] - \mathbf{x}_0 \rangle \right] = 0.
\tag{24}
$$

Thus

$$
\begin{aligned}
&\operatorname*{argmin}_{\mathbf{x}_\theta} \mathcal{L}_{\mathbf{x}_0, DSM, \phi} \\
&= \operatorname*{argmin}_{\mathbf{x}_\theta} \mathbb{E}_t \Big\{ \lambda(t) \mathbb{E}_{\mathbf{x}_0, \mathbf{x}_t} \left[ \|\mathbf{x}_\theta(\mathbf{x}_t, t, E_\phi(\mathbf{x}_0)) - \mathbb{E}[\mathbf{x}_0|\mathbf{x}_t, E_\phi(\mathbf{x}_0)]\|^2 \right] \\
&\quad + 2\lambda(t) \mathbb{E}_{\mathbf{x}_0, \mathbf{x}_t} \left[ \langle \mathbf{x}_\theta(\mathbf{x}_t, t, E_\phi(\mathbf{x}_0)) - \mathbb{E}[\mathbf{x}_0|\mathbf{x}_t, E_\phi(\mathbf{x}_0)], \mathbb{E}[\mathbf{x}_0|\mathbf{x}_t, E_\phi(\mathbf{x}_0)] - \mathbf{x}_0 \rangle \right] \Big\} \\
&= \operatorname*{argmin}_{\mathbf{x}_\theta} \mathbb{E}_t \Big\{ \lambda(t) \mathbb{E}_{\mathbf{x}_0, \mathbf{x}_t} \left[ \|\mathbf{x}_\theta(\mathbf{x}_t, t, E_\phi(\mathbf{x}_0)) - \mathbb{E}[\mathbf{x}_0|\mathbf{x}_t, E_\phi(\mathbf{x}_0)]\|^2 \right] \\
&= \mathbb{E}[\mathbf{x}_0|\mathbf{x}_t, E_\phi(\mathbf{x}_0)].
\end{aligned}
\tag{25}
$$

Substitute the minimizer of $\mathcal{L}_{\mathbf{x}_0, DSM}$ into it, we get the minimum of $\mathcal{L}_{\mathbf{x}_0, DSM}$

$$
\begin{aligned}
&\min_{\mathbf{x}_\theta} \mathcal{L}_{\mathbf{x}_0, DSM, \phi} \\
&= \min_{\mathbf{x}_\theta} \mathbb{E}_t \left\{ \lambda(t) \mathbb{E}_{\mathbf{x}_0} \mathbb{E}_{\mathbf{x}_t|\mathbf{x}_0} \left[ \|\mathbf{x}_\theta(\mathbf{x}_t, t, E_\phi(\mathbf{x}_0)) - \mathbf{x}_0\|^2 \right] \right\} \\
&= \mathbb{E}_t \left\{ \lambda(t) \mathbb{E}_{\mathbf{x}_0} \mathbb{E}_{\mathbf{x}_t|\mathbf{x}_0} \left[ \|\mathbb{E}[\mathbf{x}_0|\mathbf{x}_t, E_\phi(\mathbf{x}_0)] - \mathbf{x}_0\|^2 \right] \right\} \\
&= \mathbb{E}_t \left\{ \lambda(t) \mathbb{E}_{\mathbf{x}_t, E_\phi(\mathbf{x}_0)} \mathbb{E}_{\mathbf{x}_0|\mathbf{x}_t, E_\phi(\mathbf{x}_0)} \left[ (\mathbb{E}[\mathbf{x}_0|\mathbf{x}_t, E_\phi(\mathbf{x}_0)] - \mathbf{x}_0)^T (\mathbb{E}[\mathbf{x}_0|\mathbf{x}_t, E_\phi(\mathbf{x}_0)] - \mathbf{x}_0) \right] \right\} \\
&= \mathbb{E}_t \left\{ \lambda(t) \mathbb{E}_{\mathbf{x}_t, E_\phi(\mathbf{x}_0)} \mathbb{E}_{\mathbf{x}_0|\mathbf{x}_t, E_\phi(\mathbf{x}_0)} \left[ \operatorname{Tr}((\mathbb{E}[\mathbf{x}_0|\mathbf{x}_t, E_\phi(\mathbf{x}_0)] - \mathbf{x}_0)^T (\mathbb{E}[\mathbf{x}_0|\mathbf{x}_t, E_\phi(\mathbf{x}_0)] - \mathbf{x}_0)) \right] \right\} \\
&= \mathbb{E}_t \left\{ \lambda(t) \mathbb{E}_{\mathbf{x}_t, E_\phi(\mathbf{x}_0)} \mathbb{E}_{\mathbf{x}_0|\mathbf{x}_t, E_\phi(\mathbf{x}_0)} \left[ \operatorname{Tr}((\mathbb{E}[\mathbf{x}_0|\mathbf{x}_t, E_\phi(\mathbf{x}_0)] - \mathbf{x}_0)(\mathbb{E}[\mathbf{x}_0|\mathbf{x}_t, E_\phi(\mathbf{x}_0)] - \mathbf{x}_0)^T) \right] \right\} \\
&= \mathbb{E}_t \left\{ \lambda(t) \mathbb{E}_{\mathbf{x}_t, E_\phi(\mathbf{x}_0)} \left[ \operatorname{Tr}(\mathbb{E}_{\mathbf{x}_0|\mathbf{x}_t, E_\phi(\mathbf{x}_0)} \left[ (\mathbb{E}[\mathbf{x}_0|\mathbf{x}_t, E_\phi(\mathbf{x}_0)] - \mathbf{x}_0)(\mathbb{E}[\mathbf{x}_0|\mathbf{x}_t, E_\phi(\mathbf{x}_0)] - \mathbf{x}_0)^T) \right]) \right] \right\} \\
&= \mathbb{E}_t \left\{ \lambda(t) \mathbb{E}_{\mathbf{x}_t, E_\phi(\mathbf{x}_0)} \left[ \operatorname{Tr}(\operatorname{Cov}[\mathbf{x}_0|\mathbf{x}_t, E_\phi(\mathbf{x}_0)]) \right] \right\} \\
&= \mathbb{E}_t \left\{ \lambda(t) \mathbb{E}_{\mathbf{x}_0, \mathbf{x}_t} \left[ \operatorname{Tr}(\operatorname{Cov}[\mathbf{x}_0|\mathbf{x}_t, E_\phi(\mathbf{x}_0)]) \right] \right\} \geq 0.
\end{aligned}
\tag{26}
$$

$\square$

## A.2 LEMMAS

**Lemma 1.** $\mathbf{U}$ *and* $\mathbf{V}$ *are two square-integrable random variables.* $\mathbf{U}$ *is* $\mathcal{G}$-*measurable and* $\mathbb{E}[\mathbf{V}|\mathcal{G}] = \mathbf{0}$, *then*

$$\mathbb{E}\left[\|\mathbf{U} + \mathbf{V}\|^2\right] = \mathbb{E}\left[\|\mathbf{U}\|^2\right] + \mathbb{E}\left[\|\mathbf{V}\|^2\right]. \tag{27}$$

*Proof.*

$$\begin{aligned} &\mathbb{E}\left[\|\mathbf{U} + \mathbf{V}\|^2\right] \\ =&\mathbb{E}\left[\|\mathbf{U}\|^2\right] + \mathbb{E}\left[\|\mathbf{V}\|^2\right] + 2\mathbb{E}\left[\langle\mathbf{U}, \mathbf{V}\rangle\right], \end{aligned} \tag{28}$$

while

$$\mathbb{E}\left[\langle\mathbf{U}, \mathbf{V}\rangle\right] = \mathbb{E}\left[\mathbb{E}\left[\langle\mathbf{U}, \mathbf{V}\rangle|\mathcal{G}\right]\right] = \mathbb{E}\left[\langle\mathbf{U}, \mathbb{E}\left[\mathbf{V}|\mathcal{G}\right]\rangle\right] = 0. \tag{29}$$

$\square$

**Lemma 2.** $\mathbf{X}$ *is a random variable,* $\mathcal{F}$ *and* $\mathcal{G}$ *are two* $\sigma$-*algebras such that* $\mathcal{G} \subset \mathcal{F}$, *then we have*

$$\mathbb{E}\left[\|\mathbb{E}\left[\mathbf{X}|\mathcal{F}\right]\|^2\right] \geq \mathbb{E}\left[\|\mathbb{E}\left[\mathbf{X}|\mathcal{G}\right]\|^2\right]. \tag{30}$$

*Proof.* Let $\mathbf{U} = \mathbb{E}[\mathbf{X}|\mathcal{G}]$ and $\mathbf{V} = \mathbb{E}[\mathbf{X}|\mathcal{F}] - \mathbb{E}[\mathbf{X}|\mathcal{G}]$, $\mathbf{U}$ is $\mathcal{G}$-measurable and according to the tower property of conditional expectation

$$\mathbb{E}[\mathbf{V}|\mathcal{G}] = \mathbb{E}[\mathbb{E}[\mathbf{X}|\mathcal{F}]|\mathcal{G}] - \mathbb{E}[\mathbf{X}|\mathcal{G}] = \mathbb{E}[\mathbf{X}|\mathcal{G}] - \mathbb{E}[\mathbf{X}|\mathcal{G}] = 0. \tag{31}$$

According to lemma 1, we have

$$\mathbb{E}\left[\|\mathbb{E}\left[\mathbf{X}|\mathcal{F}\right]\|^2\right] = \mathbb{E}\left[\|\mathbb{E}\left[\mathbf{X}|\mathcal{G}\right]\|^2\right] + \mathbb{E}\left[\|\mathbb{E}\left[\mathbf{X}|\mathcal{F}\right] - \mathbb{E}\left[\mathbf{X}|\mathcal{G}\right]\|^2\right] \geq \mathbb{E}\left[\|\mathbb{E}\left[\mathbf{X}|\mathcal{G}\right]\|^2\right]. \tag{32}$$

$\square$

**Lemma 3.** *Let* $\Pi_t$ *be the set of distribution* $p(x)$ *on* $\mathbb{R}^n$ *satisfying the following condition:*

$$\mathbb{E}_p\left[\mathbf{X}\right] = \mathbf{0}, \quad \mathrm{Tr}\left(\underset{p}{\mathrm{Cov}}\left[\mathbf{X}\right]\right) = t. \tag{33}$$

*Then the n-dimensional Gaussian distribution with mean* $\mathbf{0}$ *and covariance matrix* $\Sigma = \frac{t}{n}I_n$ *is the maximum entropy distribution in* $\Pi_t$

*Proof.* We know that any probability distribution on $\mathbb{R}_n$ with finite means and finite covariances has its entropy bounded by the entropy of the n-dimensional Gaussian with the same means and covariances. Thus the maximum entropy distribution in $\mathbb{R}_n$ lies among the n-dimensional Gaussians in $\Pi_t$, which are the distributions of the form

$$p_\Sigma(\mathbf{x}) = \frac{1}{\sqrt{(2\pi)^n \det(\Sigma)}} \exp\left(-\frac{\mathbf{x}^T\Sigma^{-1}\mathbf{x}}{2}\right), \tag{34}$$

where $\Sigma$ is a positive-definite symmetric matrix with trace $t$. The entropy of $p_\Sigma$ is

$$h(p_\Sigma) = \frac{1}{2}\left(n + \log\left((2\pi)^n \det(\Sigma)\right)\right). \tag{35}$$

The arithmetic-geometric mean inequality on the eigenvalues of $\Sigma$ derives

$$\frac{1}{n}\mathrm{Tr}(\Sigma) \geq \sqrt[n]{\det(\Sigma)}. \tag{36}$$

The equality holds if and only if all the eigenvalues of $\Sigma$ are equal. Therefore

$$h(p_\Sigma) \leq \frac{n}{2}\left(1 + \log\left(\frac{2\pi t}{n}\right)\right). \tag{37}$$

Thus the n-dimensional Gaussians with mean $\mathbf{0}$ and covariance $\frac{t}{n}I_n$ is the maximum entropy distribution in $\Pi_t$. $\square$

**Lemma 4.** *The following multi-dimensional law of total variance holds*

$$\mathrm{Tr}\left(\mathrm{Cov}\left[\mathbf{Y}\right]\right) = \mathbb{E}\left[\mathrm{Tr}\left(\mathrm{Cov}\left[\mathbf{Y}|\mathbf{X}\right]\right)\right] + \mathrm{Tr}\left(\mathrm{Cov}\left[\mathbb{E}\left[\mathbf{Y}|\mathbf{X}\right]\right]\right). \tag{38}$$

*Proof.*

$$\mathrm{Tr}\left(\mathrm{Cov}\left[\mathbf{Y}\right]\right)$$
$$= \mathrm{Tr}\left(\mathbb{E}\left[\mathbf{Y}\mathbf{Y}^T\right] - \mathbb{E}\left[\mathbf{Y}\right]\mathbb{E}\left[\mathbf{Y}\right]^T\right). \tag{39}$$

Due to the property of conditional expectation, we have

$$\mathbb{E}\left[\mathbf{Y}\mathbf{Y}^T\right] = \mathbb{E}\left[\mathbb{E}\left[\mathbf{Y}\mathbf{Y}^T|\mathbf{X}\right]\right] = \mathbb{E}\left[\mathrm{Cov}\left[\mathbf{Y}|\mathbf{X}\right] + \mathbb{E}\left[\mathbf{Y}|\mathbf{X}\right]\mathbb{E}\left[\mathbf{Y}|\mathbf{X}\right]^T\right]. \tag{40}$$

$$\mathbb{E}\left[\mathbf{Y}\mathbf{Y}^T\right] - \mathbb{E}\left[\mathbf{Y}\right]\mathbb{E}\left[\mathbf{Y}\right]^T = \mathbb{E}\left[\mathrm{Cov}\left[\mathbf{Y}|\mathbf{X}\right] + \mathbb{E}\left[\mathbf{Y}|\mathbf{X}\right]\mathbb{E}\left[\mathbf{Y}|\mathbf{X}\right]^T\right] - \mathbb{E}\left[\mathbb{E}\left[\mathbf{Y}|\mathbf{X}\right]\right]\mathbb{E}\left[\mathbb{E}\left[\mathbf{Y}|\mathbf{X}\right]\right]^T$$
$$= \mathbb{E}\left[\mathrm{Cov}\left[\mathbf{Y}|\mathbf{X}\right]\right] + \mathbb{E}\left[\mathbb{E}\left[\mathbf{Y}|\mathbf{X}\right]\mathbb{E}\left[\mathbf{Y}|\mathbf{X}\right]^T\right] - \mathbb{E}\left[\mathbb{E}\left[\mathbf{Y}|\mathbf{X}\right]\right]\mathbb{E}\left[\mathbb{E}\left[\mathbf{Y}|\mathbf{X}\right]\right]^T$$
$$= \mathbb{E}\left[\mathrm{Cov}\left[\mathbf{Y}|\mathbf{X}\right]\right] + \mathrm{Cov}\left[\mathbb{E}\left[\mathbf{Y}|\mathbf{X}\right]\right]. \tag{41}$$

Take trace operation on both sides, we have

$$\mathrm{Tr}\left(\mathrm{Cov}\left[\mathbf{Y}\right]\right) = \mathbb{E}\left[\mathrm{Tr}\left(\mathrm{Cov}\left[\mathbf{Y}|\mathbf{X}\right]\right)\right] + \mathrm{Tr}\left(\mathrm{Cov}\left[\mathbb{E}\left[\mathbf{Y}|\mathbf{X}\right]\right]\right). \tag{42}$$

$\square$

### A.3 PROOF OF THEOREM 2

**Theorem 2.** *The conditioned denoising score matching objective $\mathcal{L}_{\mathbf{x}_0,DSM,\phi}$ has a smaller minimum compared with the vanilla objective:*

$$\min_{\mathbf{x}_\theta} \mathcal{L}_{\mathbf{x}_0,DSM,\phi} \leq \min_{\mathbf{x}_\theta} \mathcal{L}_{\mathbf{x}_0,DSM}. \tag{43}$$

*Proof.*

$$\min_{\mathbf{x}_\theta} \mathcal{L}_{\mathbf{x}_0,DSM} = \mathbb{E}_t\left\{\lambda(t)\mathbb{E}_{\mathbf{x}_t}\left[\mathrm{Tr}(\mathrm{Cov}[\mathbf{x}_0|\mathbf{x}_t])\right]\right\} > 0. \tag{44}$$

$$\min_{\mathbf{x}_\theta} \mathcal{L}_{\mathbf{x}_0,DSM,\phi} = \mathbb{E}_t\left\{\lambda(t)\mathbb{E}_{\mathbf{x}_0,\mathbf{x}_t}\left[\mathrm{Tr}(\mathrm{Cov}[\mathbf{x}_0|\mathbf{x}_t, E_\phi(\mathbf{x}_0)])\right]\right\} \geq 0. \tag{45}$$

It's sufficient to prove the following inequality

$$\mathbb{E}_{\mathbf{x}_0,\mathbf{x}_t}\left[\mathrm{Tr}(\mathrm{Cov}[\mathbf{x}_0|\mathbf{x}_t, E_\phi(\mathbf{x}_0)])\right] \leq \mathbb{E}_{\mathbf{x}_t}\left[\mathrm{Tr}(\mathrm{Cov}[\mathbf{x}_0|\mathbf{x}_t])\right], \tag{46}$$

which is equivalent to show

$$\mathbb{E}_{\mathbf{x}_0,\mathbf{x}_t}\left[\|\mathbb{E}[\mathbf{x}_0|\mathbf{x}_t, E_\phi(\mathbf{x}_0)] - \mathbf{x}_0\|^2\right] \leq \mathbb{E}_{\mathbf{x}_0,\mathbf{x}_t}\left[\|\mathbb{E}[\mathbf{x}_0|\mathbf{x}_t] - \mathbf{x}_0\|^2\right]. \tag{47}$$

Note that

$$\mathbb{E}_{\mathbf{x}_0,\mathbf{x}_t}\left[\|\mathbb{E}[\mathbf{x}_0|\mathbf{x}_t, E_\phi(\mathbf{x}_0)] - \mathbf{x}_0\|^2\right]$$
$$= \mathbb{E}_{\mathbf{x}_0,\mathbf{x}_t}\left[\|\mathbb{E}[\mathbf{x}_0|\mathbf{x}_t, E_\phi(\mathbf{x}_0)]\|^2\right] + \mathbb{E}_{\mathbf{x}_0,\mathbf{x}_t}\left[\|\mathbf{x}_0\|^2\right]$$
$$\quad - \mathbb{E}_{\mathbf{x}_0,\mathbf{x}_t}\left[2\langle\mathbb{E}[\mathbf{x}_0|\mathbf{x}_t, E_\phi(\mathbf{x}_0)], \mathbf{x}_0\rangle\right]$$
$$= \mathbb{E}_{\mathbf{x}_0,\mathbf{x}_t}\left[\|\mathbb{E}[\mathbf{x}_0|\mathbf{x}_t, E_\phi(\mathbf{x}_0)]\|^2\right] + \mathbb{E}_{\mathbf{x}_0,\mathbf{x}_t}\left[\|\mathbf{x}_0\|^2\right]$$
$$\quad - \mathbb{E}_{\mathbf{x}_t,E_\phi(\mathbf{x}_0)}\mathbb{E}_{\mathbf{x}_0|\mathbf{x}_t,E_\phi(\mathbf{x}_0)}\left[2\langle\mathbb{E}[\mathbf{x}_0|\mathbf{x}_t, E_\phi(\mathbf{x}_0)], \mathbf{x}_0\rangle\right]$$
$$= \mathbb{E}_{\mathbf{x}_0,\mathbf{x}_t}\left[\|\mathbb{E}[\mathbf{x}_0|\mathbf{x}_t, E_\phi(\mathbf{x}_0)]\|^2\right] + \mathbb{E}_{\mathbf{x}_0,\mathbf{x}_t}\left[\|\mathbf{x}_0\|^2\right]$$
$$\quad - 2\mathbb{E}_{\mathbf{x}_t,E_\phi(\mathbf{x}_0)}\left[\langle\mathbb{E}[\mathbf{x}_0|\mathbf{x}_t, E_\phi(\mathbf{x}_0)], \mathbb{E}[\mathbf{x}_0|\mathbf{x}_t, E_\phi(\mathbf{x}_0)]\rangle\right]$$
$$= \mathbb{E}_{\mathbf{x}_0,\mathbf{x}_t}\left[\|\mathbf{x}_0\|^2\right] - \mathbb{E}_{\mathbf{x}_0,\mathbf{x}_t}\left[\|\mathbb{E}[\mathbf{x}_0|\mathbf{x}_t, E_\phi(\mathbf{x}_0)]\|^2\right]. \tag{48}$$

Similarly, we have

$$\mathbb{E}_{\mathbf{x}_0,\mathbf{x}_t}\left[\|\mathbb{E}[\mathbf{x}_0|\mathbf{x}_t] - \mathbf{x}_0\|^2\right]$$
$$= \mathbb{E}_{\mathbf{x}_0,\mathbf{x}_t}\left[\|\mathbf{x}_0\|^2\right] - \mathbb{E}_{\mathbf{x}_0,\mathbf{x}_t}\left[\|\mathbb{E}[\mathbf{x}_0|\mathbf{x}_t]\|^2\right]. \tag{49}$$

Thus it's equivalent to proving the following inequality

$$\mathbb{E}_{\mathbf{x}_0,\mathbf{x}_t}\left[\|\mathbb{E}[\mathbf{x}_0|\mathbf{x}_t]\|^2\right] \leq \mathbb{E}_{\mathbf{x}_0,\mathbf{x}_t}\left[\|\mathbb{E}[\mathbf{x}_0|\mathbf{x}_t, E_\phi(\mathbf{x}_0)]\|^2\right]. \tag{50}$$

Note that the $\sigma$-algebra $\sigma(\mathbf{x}_t) \subset \sigma(\mathbf{x}_t, E_\phi(\mathbf{x}_0))$, according to lemma 2, the result holds. $\square$

## A.4 PROOF OF THEOREM 3

**Theorem 3.** *Suppose $\mathbf{x}_0 \in \mathbb{R}^d$, let $\mathcal{L}_{\mathbf{x}_0,DSM,\phi,t} = \mathbb{E}_{\mathbf{x}_0,\mathbf{x}_t}\left[\mathrm{Tr}(\mathrm{Cov}[\mathbf{x}_0|\mathbf{x}_t, E_\phi(\mathbf{x}_0)])\right]$ be the conditional denoising score matching loss at time $t$, and let $h(\mathbf{x}|\mathbf{y})$ be the conditional entropy of $\mathbf{x}$ given $\mathbf{y}$, then the negative logarithm of denoising score matching loss is a lower bound for the conditional mutual information between data and feature, which quantifies the shared information between $\mathbf{x}_0$ and $E_\phi(\mathbf{x}_0)$, given the knowledge of $\mathbf{x}_t$*

$$I(\mathbf{x}_0; E_\phi(\mathbf{x}_0)|\mathbf{x}_t) \geq -\log \mathcal{L}_{\mathbf{x}_0,DSM,\phi,t} + C, \quad \text{where } C = \log \frac{d}{2\pi e} + \frac{2}{d}h(\mathbf{x}_0|\mathbf{x}_t) \text{ is a constant.} \tag{51}$$

*Proof.* According to Lemma 3, we have

$$h(\mathbf{x}_0|\mathbf{x}_t = \mathbf{x}, E_\phi(\mathbf{x}_0) = \mathbf{y}) \leq \frac{d}{2}\left(1 + \log\left(\frac{2\pi \,\mathrm{Tr}\left(\mathrm{Cov}[\mathbf{x}_0|\mathbf{x}_t = \mathbf{x}, E_\phi(\mathbf{x}_0) = \mathbf{y}]\right)}{d}\right)\right). \tag{52}$$

$$\mathrm{Tr}\left(\mathrm{Cov}[\mathbf{x}_0|\mathbf{x}_t = \mathbf{x}, E_\phi(\mathbf{x}_0) = \mathbf{y}]\right) \geq \frac{d}{2\pi e}\exp\left(\frac{2h(\mathbf{x}_0|\mathbf{x}_t = \mathbf{x}, E_\phi(\mathbf{x}_0) = \mathbf{y})}{d}\right). \tag{53}$$

Taking expectation on both sides and applying Jensen's inequality (exp is a convex function)

$$\mathbb{E}_{\mathbf{x}_0,\mathbf{x}_t}\left[\mathrm{Tr}(\mathrm{Cov}[\mathbf{x}_0|\mathbf{x}_t, E_\phi(\mathbf{x}_0)])\right] \geq \frac{d}{2\pi e}\exp\left(\frac{2h(\mathbf{x}_0|\mathbf{x}_t, E_\phi(\mathbf{x}_0))}{d}\right). \tag{54}$$

Therefore, an upper bound for the conditional entropy is given by

$$h(\mathbf{x}_0|\mathbf{x}_t, E_\phi(\mathbf{x}_0)) \leq \frac{d}{2}\log\left(\frac{2\pi e}{d}\mathbb{E}_{\mathbf{x}_0,\mathbf{x}_t}\left[\mathrm{Tr}(\mathrm{Cov}[\mathbf{x}_0|\mathbf{x}_t, E_\phi(\mathbf{x}_0)])\right]\right). \tag{55}$$

We have a lower bound of the mutual information

$$\begin{aligned}
&I(\mathbf{x}_0; \mathbf{x}_t, E_\phi(\mathbf{x}_0)) \\
=&h(\mathbf{x}_0) - h(\mathbf{x}_0|\mathbf{x}_t, E_\phi(\mathbf{x}_0)) \\
\geq&h(\mathbf{x}_0) - \frac{d}{2}\log\left(\frac{2\pi e}{d}\mathbb{E}_{\mathbf{x}_0,\mathbf{x}_t}\left[\mathrm{Tr}(\mathrm{Cov}[\mathbf{x}_0|\mathbf{x}_t, E_\phi(\mathbf{x}_0)])\right]\right).
\end{aligned} \tag{56}$$

According to the chain rule of mutual information

$$I(\mathbf{x}_0; \mathbf{x}_t, E_\phi(\mathbf{x}_0)) = I(\mathbf{x}_0; \mathbf{x}_t) + I(\mathbf{x}_0; E_\phi(\mathbf{x}_0)|\mathbf{x}_t), \tag{57}$$

we have

$$\begin{aligned}
\frac{d}{2}\log\left(\frac{2\pi e}{d}\mathbb{E}_{\mathbf{x}_0,\mathbf{x}_t}\left[\mathrm{Tr}(\mathrm{Cov}[\mathbf{x}_0|\mathbf{x}_t, E_\phi(\mathbf{x}_0)])\right]\right) &\geq h(\mathbf{x}_0) - I(\mathbf{x}_0; \mathbf{x}_t) - I(\mathbf{x}_0; E_\phi(\mathbf{x}_0)|\mathbf{x}_t) \\
&\geq h(\mathbf{x}_0|\mathbf{x}_t) - I(\mathbf{x}_0; E_\phi(\mathbf{x}_0)|\mathbf{x}_t).
\end{aligned} \tag{58}$$

Thus we have proved that

$$\mathbb{E}_{\mathbf{x}_0,\mathbf{x}_t}\left[\mathrm{Tr}(\mathrm{Cov}[\mathbf{x}_0|\mathbf{x}_t, E_\phi(\mathbf{x}_0)])\right] \geq \frac{d}{2\pi e}\exp\left(\frac{2}{d}h(\mathbf{x}_0|\mathbf{x}_t)\right)\exp\left(-I(\mathbf{x}_0; E_\phi(\mathbf{x}_0)|\mathbf{x}_t)\right). \tag{59}$$

But we have

$$\mathcal{L}_{\mathbf{x}_0,DSM,\phi,t} \geq \mathbb{E}_{\mathbf{x}_0,\mathbf{x}_t}\left[\mathrm{Tr}(\mathrm{Cov}[\mathbf{x}_0|\mathbf{x}_t, E_\phi(\mathbf{x}_0)])\right]. \tag{60}$$

Thus

$$\mathcal{L}_{\mathbf{x}_0,DSM,\phi,t} \geq \frac{d}{2\pi e}\exp\left(\frac{2}{d}h(\mathbf{x}_0|\mathbf{x}_t)\right)\exp\left(-I(\mathbf{x}_0; E_\phi(\mathbf{x}_0)|\mathbf{x}_t)\right). \tag{61}$$

We get the result after rearranging the above equation

$$I(\mathbf{x}_0; E_\phi(\mathbf{x}_0)|\mathbf{x}_t) \geq -\log \mathcal{L}_{\mathbf{x}_0,DSM,\phi,t} + \log \frac{d}{2\pi e} + \frac{2}{d}h(\mathbf{x}_0|\mathbf{x}_t). \tag{62}$$

$\square$

## B  ARCHITECTURE OF GRAPH-UNET

As illustrated on the right side of fig. 1, our decoder adopts a UNet-like architecture, comprising a contracting path (left side) and an expansive path (right side). However, since up-sampling and down-sampling operations cannot be directly applied to graph data, we instead represent the granularity of modeling through dimensional reduction and expansion. Specifically, due to the requirement of the diffusion model that the input and output dimensions match the original feature dimensions, we introduce additional input and output layers to perform dimensional mappings. In the contracting path, repeated dimensional reduction is performed using either GNN layers or MLP layers, depending on different task types, which halves the number of hidden dimensions at each step. In the expansive path, dimensional expansion is repeated, but before each mapping, the hidden state of the corresponding contracting path with the same dimension is added via skip connections, which differs from the original UNet's concatenation.

It is also important to note that, in addition to the noisy data $\mathbf{x}_t$, the decoder also receives the condition $\mathbf{z}$ and time $t$ as inputs. We encode the time information using two linear layers with SiLU activation (Elfwing et al., 2018), and employ positional encoding to enable the model to distinguish temporal order. Furthermore, a key challenge is how to fuse $\mathbf{x}_t$, $\mathbf{z}$, and $t$. Based on experimental results, the optimal approach for node-level tasks is to directly sum these three components after encoding, as shown below:

$$\mathbf{h}^{(l+1)} = \mathbf{h}^{(l)} + \texttt{MLP}_t(t) + \texttt{MLP}_z(\mathbf{z}) \tag{63}$$

where $\texttt{MLP}_t(\cdot)$ and $\texttt{MLP}_z(\cdot)$ are both MLP layer to achieve dimensional mapping.

For graph-level tasks, we follow the approach commonly used in the field of computer vision, utilizing Adaptive Normalization layers (Dhariwal & Nichol, 2021; Hudson et al., 2024) to fuse the three components:

$$\mathbf{h}^{(l+1)} = \texttt{AdaNorm}(\mathbf{h}^{(l)}, \mathbf{z}, t) = \mathbf{z}_s(t_s \texttt{LayerNorm}(\mathbf{h}^{(l)}) + t_b) + \mathbf{z}_b \tag{64}$$

where $(t_s, t_b)$ and $(\mathbf{z}_s, \mathbf{z}_b)$ are both obtained by linear projection.

## C  HYPER-PARAMTER CONFIGURATIONS

Table 4: Hyper-parameter configurations for node classification datasets.

| | Dataset | Cora | CiteSeer | PubMed | Ogbn-arxiv | Computer | Photo |
|---|---|---|---|---|---|---|---|
| | feat_drop | 0.3 | 0.4 | 0.2 | 0.1 | 0.4 | 0.1 |
| | att_drop | 0.1 | 0.2 | 0.2 | 0.2 | 0.2 | 0.3 |
| | num_head | 4 | 4 | 2 | 2 | 2 | 4 |
| Hyper-parameters | num_hidden | 1024 | 1024 | 1024 | 256 | 512 | 512 |
| | learning_rate | 1e-4 | 1e-4 | 1e-4 | 1e-3 | 1e-4 | 3e-4 |
| | mask_ratio | 0.7 | 0.7 | 0.7 | 0.7 | 0.7 | 0.7 |
| | noise_schedule | sigmoid | sigmoid | sigmoid | inverted | quad | sigmoid |
| | optimizer | Adam | Adam | Adam | Adam | Adam | Adam |

Table 5: Hyper-parameter configurations for graph classification datasets.

| | Dataset | IMDB-B | IMDB-M | PROTEINS | COLLAB | MUTAG |
|---|---|---|---|---|---|---|
| | feat_drop | 0.2 | 0.2 | 0.2 | 0.2 | 0.2 |
| | num_hidden | 512 | 512 | 512 | 256 | 256 |
| Hyper-parameters | learning_rate | 1.5e-4 | 1.5e-4 | 1.5e-4 | 1.5e-4 | 1e-4 |
| | mask_ratio | 0 | 0 | 0 | 0.3 | 0 |
| | noise_schedule | sigmoid | sigmoid | sigmoid | sigmoid | sigmoid |
| | optimizer | AdamW | AdamW | AdamW | AdamW | AdamW |

# D  ADDITIONAL EXPERIMENTS

## D.1  ABLATION STUDY ON ENCODER BACKBONE

To evaluate how much impact the choice of encoder has on the performance of `Graffe` and other baselines, we conduct ablation studies on the encoder backbone using three classic datasets: Cora, Citeseer, and PubMed. We chose GRACE (Zhu et al., 2021) and CCA-SSG (Zhang et al., 2021) as baselines for contrastive learning and GraphMAE (Hou et al., 2022), MaskGAE (Li et al., 2023), and Bandana (Zhao et al., 2024) as baselines for the MAE family. The experimental results are shown in Table 6.

Table 6: Ablation study on different encoder design.

| Method | Cora | | Citeseer | | Pubmed | |
|---|---|---|---|---|---|---|
| | GCN | GAT | GCN | GAT | GCN | GAT |
| GRACE | 81.9±0.4 | 81.0±0.6 | 71.2±0.5 | 71.5±0.5 | 80.6±0.4 | 78.9±0.2 |
| GraphMAE | 82.5±0.5 | 84.2±0.4 | 72.6±0.6 | 73.4±0.4 | 80.9±0.2 | 81.1±0.4 |
| CCA-SSG | 84.0±0.4 | 82.7±0.6 | 73.1±0.3 | 72.3±0.6 | 81.0±0.5 | 80.7±0.9 |
| *MaskGAE$_{edge}$ | 83.8±0.3 | 82.0±0.1 | 72.9±0.2 | 72.0±0.4 | 82.7±0.3 | 81.2±0.1 |
| Bandana | 84.5±0.3 | 83.1±0.6 | 73.6±0.2 | 73.7±0.5 | 83.7±0.5 | 81.5±0.8 |
| **Graffe** | 83.2±0.5 | 84.8±0.4 | 73.2±0.2 | 74.3±0.4 | 80.5±0.4 | 81.0±0.6 |

*Results marked with * are taken from the original literature.*

The results show significant performance declines for many methods when substituting GCN for GAT, such as CCA-SSG, MaskGAE, and Bandana on Cora and Citeseer dataset, which also aligns with observations in Table 5 of MaskGAE (Li et al., 2023). In contrast, for GraphMAE and `Graffe`, switching their GAT backbones to GCN cause a noticeable drop in performance. We believe different SSL methods have distinct encoder preferences and using GAT or GCN as the encoder in graph SSL is not universally optimal.

## D.2  ABLATION STUDY ON GRAPH-UNET BACKBONE

As mentioned in Appendix B, we chose the Unet structure because it can capture information at different granularities while strictly ensuring input-output dimensional consistency. During our early exploration, we also tested using a simple MLP or GNN as the decoder. The experimental results on Cora, Photo, and IMDB-B datasets are shown in Table 7. It is worth noting that the GNN decoder adopts the same architecture as the encoder: GAT for node-level tasks and GIN for graph-level tasks.

Table 7: Ablation study on different decoder design.

| Decoder | Cora | Computer | IMDB-B |
|---|---|---|---|
| MLP | 82.6±0.5 | 89.1±0.1 | 75.0±0.6 |
| GNN (GAT/GIN) | 80.2±0.3 | 88.1±0.1 | 74.5±0.5 |
| Graph-Unet | 84.8±0.4 | 91.3±0.2 | 76.2±0.2 |

We can observe that using either an MLP or GNN as the decoder results in significantly poorer performance compared to the Graph-Unet. Moreover, for node-level tasks, employing a GNN as the decoder leads to a substantial performance drop. This observation aligns with our analysis in Section 4.2, where we note that GNNs can cause interference among nodes due to varying degrees of noise introduced during the diffusion process.

# E   DISCUSSIONS

**Intuitive guide for choosing masking ratio** $m$**.**   As shown in Appendix C, although the optimal mask ratio differs across datasets, there are clear trends across different tasks. For instance, a larger mask ratio generally yields better results in node classification, while the opposite is true for graph classification. We hypothesize that this may be due to the combined effect of the graph characteristics and diffusion representation learning. Here we provide some intuitive understanding. In graph classification tasks, where graphs are typically small and have simpler connectivity, a small mask ratio is suggested to avoid significant information loss. Conversely, in node classification tasks, where there are more nodes and more complex connections, a large mask ratio is suggested since overly detailed modeling can cause the model to become overly focused on intricate information. We suggest that when selecting the mask ratio, one should first assess the characteristics of the graph and then determine an appropriate candidate for the mask ratio accordingly.

**Limitations and future work**   Despite the significant contributions of this study to the understanding of DRL and significant performance on graph tasks, there are certain limitations that should be acknowledged to provide a comprehensive perspective. The proposed Diff-InfoMax principle involves a weighting function over time. How to dynamically adjust the weighting function over different data and tasks remains an unsolved problem. Additionally, methods to optimize alternative variational lower bounds of the Diff-InfoMax principle are left for future exploration. From an empirical perspective, we believe that the structure, or the non-Euclidean nature of graphs, is crucial information for graph representation learning. Therefore, an intriguing question remains regarding the deeper understanding of explicitly incorporating structural modeling into diffusion representation learning, which is a highly non-trivial task. Moreover, our method does not have an advantage in terms of time efficiency. Improvements in training speed and further refinements in model design are left as directions for future research.

