# OpenReview forum: "Graffe: Graph Representation Learning Enabled via Diffusion Probabilistic Models"
_ICLR.cc/2025/Conference — Submitted to ICLR 2025_

### Official Review · Reviewer_36MZ · 2024-10-30

**Soundness:** 3
**Presentation:** 2
**Contribution:** 3
**Rating:** 6
**Confidence:** 3

**Summary:**

Graph representation learning has been a critical problem and is commonly addressed by graph neural networks. This paper explores diffusion probabilistic models (DPMs) on graph learning and proposes the Graffe model. Theoretically, the theoretical foundations of Graffe are proved to be the objective of maximizing the conditional mutual information between data and presentations. Empirically, Graffe achieves superior performance on 9 out of 11 datasets on node and graph classification tasks.

**Strengths:**

1. A necessary theoretical foundation is provided for diffusion models to presentation learning. The lower bound is derived as the objective for model training.

2. Critical claims and designs are well justified with experimental results, e.g., Figure 2 and Figure 3.

3. Experimental results support the effectiveness of the proposed Graffe model.

**Weaknesses:**

1. As claimed, the aim of the diffusion model for representation learning is to maximize the conditional mutual information between data and its presentation. However, in Equation (8), it seems to replace presentations with labels. Please justify the motivation.

2. Some notations are not properly defined before usage. For example, f(t), g(t), λ(t) in Section 2 and Operator Tr, Cov in Section 3. Those undefined notations impact the comprehension of the content.

3.  The foundation of the diffusion process for representation learning is to maximize the mutual information of data and presentations. However, this design has been discussed for a long time in the design of GNN, such as [1][2]. Could the authors please justify the fundamental differences if any?

[1] Learning with Local and Global Consistency, NIPS, 2003

[2] Interpreting and Unifying Graph Neural Networks with An Optimization Framework, WWW, 2021

4. Polynomial spectral GNNs have demonstrated advantages in graph representation learning. Please justify the advantages of Graffe, a diffusion model over those polynomial GNNs.

5. Several state-of-the-art models for node classification are missing in the experiments, e.g., [1][2][3]. Please include them for comparison in the experiments.

[1] Graph Neural Networks with Learnable and Optimal Polynomial Bases, ICML, 2023

[2] How Universal Polynomial Bases Enhance Spectral Graph Neural Networks, ICML, 2024

[3] Specformer: Spectral graph neural networks meet transformers. ICLR, 2023

**Questions:**

Please refer to the weakness part.

---

> ### Author Response · Authors · 2024-11-20
> **Response to Reviewer 36MZ**
>
> > **1. As claimed, the aim of the diffusion model for representation learning is to maximize the conditional mutual information between data and its presentation. However, in Equation (8), it seems to replace presentations with labels. Please justify the motivation.**
>
> Thanks for your feedback. We would like to clarify that the wording 'conditioned denoising score matching' in Theorem 8 refers to (representation) conditioned denoising score matching. $E_\phi(x_0)$ in Equation (8) of Theorem 1 represents a general data representation. The results established in Theorems 1, 2, and 3 are applicable to this general representation, $E_\phi(x_0)$. In line 156, we regard the class label as a 1-dimensional feature.
>
>
>
> > **2. Some notations are not properly defined before usage. For example, f(t), g(t), λ(t) in Section 2 and Operator Tr, Cov in Section 3. Those undefined notations impact the comprehension of the content.**
>
> We apologize for any confusion caused by the undefined notations. The functions $f(t)$ and $g(t)$ are scalar functions, defined as $f, g: \mathbb{R} \rightarrow \mathbb{R}$, which determine the forward noising process of the diffusion model, such that $x_t | x_0 \sim \mathcal{N}(\alpha_t x_0, \sigma_t^2 I)$. These functions are closely related to the parameters $\alpha_t$ and $\sigma_t$. Additionally, $Tr$ denotes the trace of a matrix, while $Cov$ represents the covariance matrix between random variables. We revised the paper to improve clarity and comprehension.
>
>
>
> > **3. The foundation of the diffusion process for representation learning is to maximize the mutual information of data and presentations. However, this design has been discussed for a long time in the design of GNN, such as[1][2]. Could the authors please justify the fundamental differences if any?**
>
> Thanks for your comments. After carefully reviewing the related works you mentioned, we found that work one [1] approaches the relationship between representation learning and classification tasks from a semi-supervised learning perspective, whereas our work focuses on self-supervised representation learning. Work 2 [2], on the other hand, provides theoretical insights and unified framework of different GNN architectures, while we focus on learning strategy as mentioned in **General Response**. As such, our work is not directly related to either of these references.
>
> We acknowledge that many theoretical contributions in representation learning aim to elucidate the relationship between information theory and the proposed methods. However, establishing a connection between diffusion representation learning and information theory remains unexplored, and our work is the first to theoretically establish this feasibility. We hope this explanation helps the reviewer better understand the theoretical contributions of our work.
>
>
>
> > **4. Polynomial spectral GNNs have demonstrated advantages in graph representation learning. Please justify the advantages of Graffe, a diffusion model over those polynomial GNNs.**
> >
> > **5. Several state-of-the-art models for node classification are missing in the experiments, e.g., [1, 2, 3]. Please include them for comparison in the experiments.**
>
> Since the concerns raised in these two questions are similar, we have decided to address them together. First, we would like to thank the reviewer for introducing us to these interesting works. However, we believe there are some misunderstanding regarding the research problem we are addressing. We kindly invite the reviewer to refer to our explanations in the **General Response** section. Below, we provide point-to-point answers to these two questions.
>
> - While Polynomial Spectral GNNs show promising results in graph representation learning, they fall under the category of GNN architecture design. Our work focuses on learning strategy design, i.e., designing more effective self-supervised strategies while keeping the backbone model fixed. Therefore, we regret that we cannot provide a direct comparison, as we believe **these are two orthogonal research directions**.
>
> - Although the works mentioned by the reviewer do not address the same research problem as our method, they can indeed be combined to achieve better performance on graph-related tasks. Therefore, we conducted additional experiments using SpecFormer as the encoder backbone and applying the Graffe method for representation learning.
>
>   |                       | Cora         | Citeseer     |
>   | --------------------- | ------------ | ------------ |
>   | **GAT+Graffe**        | 84.8$\pm$0.4 | 74.3$\pm$0.4 |
>   | **Specformer+Graffe** | 85.5$\pm$0.6 | 75.6$\pm$0.5 |
>
>   As shown in our results, this combination does lead to performance improvements, which we agree is valuable for practical applications. However, this falls beyond the scope of the discussions in our paper. We hope our explanation provides the reviewer a clearer understanding of our work.

---

> > ### Comment · Reviewer_36MZ · 2024-11-26
> >
> > Thanks for the responses. My concerns are basically addressed. I updated the score accordingly.

---

> > > ### Author Response · Authors · 2024-11-26
> > >
> > > Thank you for taking time to review our responses and for updating the score! We sincerely appreciate your thoughtful feedback, which has greatly helped us improve the quality of our manuscript.

---

### Official Review · Reviewer_W23D · 2024-11-02

**Soundness:** 3
**Presentation:** 3
**Contribution:** 3
**Rating:** 6
**Confidence:** 4

**Summary:**

Graph representation learning via diffusion-based approaches which reaches Sota performance.

**Strengths:**

The writing and visualization are pretty clear and the paper also includes core ablations. The usage of graph U-Net is well-motivated and clarified in the paper. The performance seems to be consistent in the ablations in 5.4. The theoretical insights about conditioning supports better the encoder part.

**Weaknesses:**

1. Some key insights are missing from the analysis:
  * The effect of mask ratio seems to be very different for different datasets, which will lead to more tuning, as mentioned in the limitations. However, the insight about the impact of mask ratio is also missing in the discussions, and an intuitive guide for choosing this ratio would be helpful.
  * It is very intriguing that A reconstruction seems to only deteriorate the performance as mentioned in the 1st part of 5.4, since A contains richer information than node features for graphs without explicit node features. The insight about this is also missing.
2. While spectra are a key in related work, this is no further analysis in the experiments and in methods. The related sentence in Line 84 of the Introduction lacks supportive evidence or explanation.

**Questions:**

One question I have is about the difference between the 'encoder' and 'decoder' part, where 'encoder' functions as a condition for the decoder. The mechanism of 'encoder' by masking nodes is very similar to another diffusion process which absorbs nodes sequentially until all nodes become noisy. So the process seems to be somehow a denoising process given 2 types of noising trajectory (one with t being dynamic, and another with t being fixed as the mask ratio). Would the performance change if the encoder/decoder designs switch, out of curiosity? Have you considered using both dynamic 't' as the mask ratio in the 1st part as well?

---

> ### Author Response · Authors · 2024-11-20
> **Response to Reviewer W23D (1/2)**
>
> > **1. The effect of mask ratio seems to be very different for different datasets, which will lead to more tuning, as mentioned in the limitations. However, the insight about the impact of mask ratio is also missing in the discussions, and an intuitive guide for choosing this ratio would be helpful.**
>
> Thanks for your constructive feedback. As shown in `Table 4` of the Appendix, although the optimal mask ratio differs across datasets, there are clear trends across different tasks. For instance, a larger mask ratio generally yields better results in node classification, while the opposite is true for graph classification. We hypothesize that this may be due to the combined effect of the graph characteristics and diffusion representation learning.
>
> Here we provide some intuitive understanding. In graph classification tasks, where graphs are typically small and have simpler connectivity, a small mask ratio is suggested to avoid significant information loss. Conversely, in node classification tasks, where there are more nodes and more complex connections,  a large mask ratio is suggested since overly detailed modeling can cause the model to become overly focused on intricate information.
>
> We suggest that when selecting the mask ratio, one should first assess the characteristics of the graph and then determine an appropriate candidate for the mask ratio accordingly. In the `Appendix E` of the revised manuscript, we have added a related discussion and provided some intuitive guidance.
>
>
>
> > **2. It is very intriguing that A reconstruction seems to only deteriorate the performance as mentioned in the 1st part of 5.4, since A contains richer information than node features for graphs without explicit node features. The insight about this is also missing.**
>
> Thank you for pointing this out. We acknowledge that the experimental results are indeed counterintuitive, but we would like to clarify that the performance decline when incorporating structural information does not indicate that structural information is not important. As we expressed in our response to Reviewer TYZT's second question, introducing more explicit structural modeling into diffusion representation learning is highly non-trivial, as simple applications fail to yield satisfactory performance. We believe that the structure, or the non-Euclidean nature of graphs, is crucial information. However, the main contribution of our work lies in demonstrating the feasibility of DRL for graph representation learning from both theoretical and experimental perspectives. We hope to treat the more complex design of incorporating structural information as a future direction. In the revised manuscript's `Appendix E`, we have added the relevant discussion, and we hope this addresses your concern.
>
>
>
> > **3. While spectra are a key in related work, this is no further analysis in the experiments and in methods. The related sentence in Line 84 of the Introduction lacks supportive evidence or explanation.**
>
> We apologize for any confusion caused. The related sentence in Line 84 was to summarize our claims in `Section 3.3` of the manuscript. Our intention was to introduce the connection between the diffusion model and the frequency domain of graph features, rather than the more familiar spectrum domain of graph topology. We have revised the wording in the `Introduction` and provided a clearer explanation.

---

> ### Author Response · Authors · 2024-11-20
> **Response to Reviewer W23D (2/2)**
>
> > **[Q1] One question I have is about the difference between the 'encoder' and 'decoder' part, where 'encoder' functions as a condition for the decoder. The mechanism of 'encoder' by masking nodes is very similar to another diffusion process which absorbs nodes sequentially until all nodes become noisy. So the process seems to be somehow a denoising process given 2 types of noising trajectory (one with t being dynamic, and another with t being fixed as the mask ratio). Would the performance change if the encoder/decoder designs switch, out of curiosity? Have you considered using both dynamic 't' as the mask ratio in the 1st part as well?**
>
> We appreciate your keen intuition and valuable comments, and indeed, this is a very interesting issue. As you point out, the masking of nodes is similar to the forward process of absorbing/masked discrete state diffusion. However, we do not aim to generate the nodes in a discrete diffusion fashion.
>
> - Let's discuss the second question regarding the use of dynamic $\mathbf{t}$ in the encoder. In fact, we had attempted this approach during the strategy design phase. However, unlike your understanding from the perspective of noising trajectory, our initial goal was to use a dynamic mask ratio to coordinate the training of the encoder and decoder. Intuitively, during the early stages of training, we aimed for the denoising decoder to gain more training and improve its unconditional reconstruction ability. While in the later stages, once the decoder was of high quality, we shifted focus to training the encoder to squeeze the capability of representation learning. Therefore, we set the mask ratio to an inverted linear schedule. However, the experimental results did not show a significant performance improvement, so we abandoned this approach. Nonetheless, we believe there are still valuable insights to explore in this direction.
>
> - Regarding the first issue, the switch in noising trajectory, we followed your suggestion and conducted relevant experiments on Cora, Computer, and IMDB-B datasets. The experimental results are as follows:
>
>   |                             | Cora         | Computer     | IMDB-B       |
>   | --------------------------- | ------------ | ------------ | ------------ |
>   | Switched noising trajectory | 81.8$\pm$0.8 | 85.3$\pm$0.4 | 71.8$\pm$0.3 |
>   | Graffe                      | 84.8$\pm$0.4 | 91.3$\pm$0.2 | 76.2$\pm$0.2 |
>
> ​	As can be seen, there is a significant performance drop, and it seems somewhat weird that using the same $\mathbf{t}$ for the diffusion process, which causes the model to degrade into a denoising auto-encoder (DAE).

---

> > ### Comment · Reviewer_W23D · 2024-11-24
> >
> > Thank you for your reply and it has addressed most of my questions. The only part that remains unclear is the A reconstruction. However, I understand that the focus of this work is on representation learning rather than reconstruction itself.
> > I maintain a positive score of 6 for this manuscript.

---

> > > ### Author Response · Authors · 2024-11-25
> > >
> > > Thank you for your response! We are glad to hear that most of your questions have been addressed and appreciate your understanding regarding the focus of our work.

---

### Official Review · Reviewer_TYZT · 2024-11-03

**Soundness:** 3
**Presentation:** 3
**Contribution:** 3
**Rating:** 6
**Confidence:** 4

**Summary:**

This paper proposes Graffe, a method that applies Diffusion models, currently the most popular deep generative models, to self-supervised representation learning on graphs. It first uses a graph encoder to obtain a compact representation of a graph/node, then uses a conditional generative model to recover node/graph features based on this representation. Through theoretical analysis, the paper proves that diffusion models can maximize the conditional mutual information between the data and its embeddings. In terms of experimental results, Graffe delivers superior performance compared to previous self-supervised learning models on 9 out of 11 node classification and graph classification datasets.

**Strengths:**

- I really like the idea proposed in this paper. I am quite familiar with graph self-supervised learning and diffusion models (as well as the Infomax principle), and this paper manages to combine these two seemingly distant topics in a very promising and exciting way.
- The paper provides substantial theoretical proofs and analytical experiments (such as Figure 2), demonstrating the relationships between (conditional) diffusion models, mutual information, and effective data representations.
- The experimental results are very comprehensive. First, it includes both self-supervised node classification datasets and self-supervised graph classification datasets. Second, based on my experience, Graffe achieves state-of-the-art performance on many benchmarks, which is a very impressive result.

**Weaknesses:**

- Since DDPM can be understood as a special type of variational autoencoder, this paper could also be interpreted as a special VAE-based self-supervised method.
- In Section 1, the paper mentions the challenge in generalizing the representation learning power of diffusion models on graph data, with one being the non-Euclidean nature of graph data. However, besides using GNN models as encoder and decoder, Graffe doesn't seem to make additional considerations for this challenge.
- Although the paper provides very detailed theoretical analysis, its conclusions seem relatively trivial: First, since representation is inherently a function of the input, Theorem 2 is obvious. Second, the InfoMax conclusion doesn't explain why the learned representations would be better, because as mentioned in the paper, identity mapping would be the encoding that maximizes Mutual Information. Third, there is a significant gap between theoretical analysis and practical application. The theoretical analysis seems to be aimed at i.i.d. data, where both the encoder's input and decoder's target are x. However, for graphs, the encoder's input consists of two parts (node features and graph structure), while the target is only the node features.

**Questions:**

1. I want to know how much impact the choice of Encoder has on Graffe's performance. As is well known, many node-level self-supervised models (such as most contrastive learning models) use the most basic two-layer GCN model. Is using GAT unfair for Graffe (I know some other Graph MAE models also use GAT)? How would using a GCN model affect Graffe's performance? What would happen if contrastive learning methods also used GAT models?

2. How important is the Unet structure for the Decoder? What impact would using regular MLPs or GNNs have on performance?

3. How efficient is Graffe? For example, how does it compare in terms of training time with other contrastive methods or MAE methods?

---

> ### Author Response · Authors · 2024-11-20
> **Response to Reviewer TYZT (1/3)**
>
> > **1. Since DDPM can be understood as a special type of variational autoencoder, this paper could also be interpreted as a special VAE-based self-supervised method.**
>
> Thank you for the insightful suggestion. Indeed, DDPM can be interpreted as a hierarchical VAE[1], and viewing our method through the lens of VAE-based representation learning [2, 3] opens up an interesting perspective. We believe this interpretation could provide valuable insights and further enrich our understanding of the results.
>
> [1] Variational Diffusion Models. NeurIPS 2021
>
> [2] Learning deep representations by mutual information estimation and maximization. ICLR 2019
>
> [3] Improving VAE-based Representation Learning.
>
>
>
> > **2. In Section 1, the paper mentions the challenge in generalizing the representation learning power of diffusion models on graph data, with one being the non-Euclidean nature of graph data. However, besides using GNN models as encoder and decoder, Graffe doesn't seem to make additional considerations for this challenge.**
>
> Thanks for your detailed review and constructive comments. The non-Euclidean nature of graphs has been a key focus for us when designing the Graffe framework, especially in finding effective ways to handle graph topology. However, during our exploration, we found that the reality was far more complex than we expected. Here are some early attempts we made during this exploration process:
>
> 1. **Reconstructing the Adjacency Matrix**: As mentioned in `Section 4.2` of our manuscript, we tried to incorporate adjacency matrix reconstruction into the diffusion process to better capture structural information. However, empirical analysis shows some counterintuitive results, and this approach significantly increased computational costs.
>
> 2. **Reconstructing Structure-Related Features**: Building on the previous idea, we also tried introducing structure-related features into node attributes, such as node2vec [1] and eigenvector positional encoding [2]. We thought this could help the enhanced features $\mathbf{\hat{X}}$ capture structure information while keeping computation manageable. The results are listed below:
>
>    |                                            | Cora | Computer |
>    | ------------------------------------------ | ---- | -------- |
>    | $\mathbf{X}$ Recons.                       | 84.8 | 91.3     |
>    | $\mathbf{A}$ Recons.                       | 77.6 | 86.2     |
>    | $\mathbf{\hat{X}}$ Recons. (Node2vec)      | 81.6 | 88.2     |
>    | $\mathbf{\hat{X}}$ Recons. (eigenvecor PE) | 81.9 | 89.0     |
>
>    We find that it still fell short of the performance achieved by reconstructing node features only. This indicates that such strategies remain suboptimal.
>
> These early explorations highlight that integrating structural information into diffusion representation learning is a highly non-trivial task. As a result, we decided to stick with the simple yet effective feature reconstruction approach for our final design.
>
> However, we want to emphasize that the primary contribution of our work is demonstrating the feasibility of DPMs for graph SSL, with additional theoretical support and empirical insights. We leave it as future work to explore how to better incorporate structural information in diffusion representation learning. We hope this response addresses your concern.
>
> [1] node2vec: Scalable Feature Learning for Networks. KDD 2016.
>
> [2] Rethinking Graph Transformers with Spectral Attention. NeurIPS 2021.

---

> ### Author Response · Authors · 2024-11-20
> **Response to Reviewer TYZT (2/3)**
>
> > **3. Although the paper provides very detailed theoretical analysis, its conclusions seem relatively trivial: First, since representation is inherently a function of the input, Theorem 2 is obvious. Second, the InfoMax conclusion doesn't explain why the learned representations would be better, because as mentioned in the paper, identity mapping would be the encoding that maximizes Mutual Information. Third, there is a significant gap between theoretical analysis and practical application. The theoretical analysis seems to be aimed at i.i.d. data, where both the encoder's input and decoder's target are x. However, for graphs, the encoder's input consists of two parts (node features and graph structure), while the target is only the node features.**
>
> Thank you for the thoughtful feedback. We respectfully disagree that the conclusions seem relatively trivial. We acknowledge that Theorem 2 may appear intuitively obvious, as prior diffusion-based representation learning methods have also relied on this intuition. However, our intention with Theorem 2 was to provide a rigorous theoretical justification for this widely accepted notion. Additionally, to the best of our knowledge, the findings presented in Theorem 3 are **novel and have not been previously documented**.
>
> Regarding the InfoMax concern, we agree that identity mapping would maximize mutual information by retaining all the information from the input. To mitigate the risk of the model learning such a shortcut, we specifically designed the encoder architecture and introduced masking as a regularization to constrain the encoder, thereby encouraging more meaningful representations.
>
> For the third problem, as shown in `Table 3`, we experimented with different decoder targets and empirically found that incorporating adjacency matrix reconstruction into the diffusion process negatively impacts performance. If we treat the node features as the data and the graph structure as additional conditional information, this observation aligns with our theoretical analysis under conditional settings.
>
>
>
> > **[Q1] I want to know how much impact the choice of Encoder has on Graffe's performance. As is well known, many node-level self-supervised models (such as most contrastive learning models) use the most basic two-layer GCN model. Is using GAT unfair for Graffe (I know some other Graph MAE models also use GAT)? How would using a GCN model affect Graffe's performance? What would happen if contrastive learning methods also used GAT models?**
>
> Thank you for pointing out this issue. As per your suggestion, we conducted ablation studies on the encoder backbone using three classic datasets: Cora, Citeseer, and PubMed. We select GRACE and GraphMAE as baselines for contrastive learning and MAE, respectively. Below are the experimental results:
>
> |                    | Cora         | Citeseer     | Pubmed       |
> | ------------------ | ------------ | ------------ | ------------ |
> | **GRACE (GCN)**    | 81.9$\pm$0.4 | 71.2$\pm$0.5 | 80.6$\pm$0.4 |
> | **GRACE (GAT)**    | 81.0$\pm$0.6 | 71.5$\pm$0.5 | 78.9$\pm$0.2 |
> | **GraphMAE (GCN)** | 82.5$\pm$0.5 | 72.6$\pm$0.6 | 80.9$\pm$0.2 |
> | **GraphMAE (GAT)** | 84.2$\pm$0.4 | 73.4$\pm$0.4 | 81.1$\pm$0.4 |
> | **Graffe (GCN)**   | 83.2$\pm$0.5 | 73.2$\pm$0.2 | 80.5$\pm$0.4 |
> | **Graffe (GAT)**   | 84.8$\pm$0.4 | 74.3$\pm$0.4 | 81.0$\pm$0.6 |
>
> It can be observed that for GRACE, replacing GCN with GAT led to varying results across different datasets, with some improvements and some declines. However, for GraphMAE and Graffe, replacing GAT with GCN resulted in a obvious performance drop. Despite this, **Graffe still maintained its relative advantage** when using the same backbone. It is worth noting that the backbone may influence the choice of optimal hyperparameters, and thus the reproduced results here might not reflect the upper limit of each backbone’s performance. However, we want to emphasize that our choice of GAT as the backbone aligns with recent representation learning efforts, particularly those in the MAE family. We have added the above discussion in our `Appendix D.1` for your references.

---

> ### Author Response · Authors · 2024-11-20
> **Response to Reviewer TYZT (3/3)**
>
> > **[Q2] How important is the Unet structure for the Decoder? What impact would using regular MLPs or GNNs have on performance?**
>
> Thank you for pointing this out. The history of the diffusion model can be traced back to 2015[1]. However, the empirical success in DDPM[2] and Score-SDE[3] (and former NCSN[4]) relies on the UNet architecture. In [5], the authors also comment that they have tried ResNet backbone in early experiments but find its performance falls short of UNet. As mentioned in Appendix B, we chose the UNet structure because it can capture information at different granularities while strictly ensuring input-output dimensional consistency. During our early exploration, we also tested using a simple MLP or GNN as the decoder. Below are the experimental results on Cora, Photo, and IMDB-B datasets. It is worth noting that the GNN decoder adopts the same architecture as the encoder: GAT for node-level tasks and GIN for graph-level tasks.
>
> | Decoder       | Cora         | Computer     | IMDB-B       |
> | ------------- | ------------ | ------------ | ------------ |
> | MLP           | 82.6$\pm$0.5 | 89.1$\pm$0.1 | 75.0$\pm$0.6 |
> | GNN (GAT/GIN) | 80.2$\pm$0.3 | 88.1$\pm$0.1 | 74.5$\pm$0.5 |
> | Graph-Unet    | 84.8$\pm$0.4 | 91.3$\pm$0.2 | 76.2$\pm$0.2 |
>
> We can observe that using either an MLP or GNN as the decoder results in **significantly poorer performance** compared to the Graph-Unet. Moreover, for node-level tasks, employing a GNN as the decoder leads to a substantial performance drop. This observation aligns with our analysis in `Section 4.2`, where we note that GNNs can cause interference among nodes due to varying degrees of noise introduced during the diffusion process. We have also added the above discussion in our `Appendix D.2` for your references.
>
> [1] Deep Unsupervised Learning using Nonequilibrium Thermodynamics.
>
> [2] Denoising Diffusion Probabilistic Models.
>
> [3] Score-Based Generative Modeling through Stochastic Differential Equations.
>
> [4] Generative Modeling by Estimating Gradients of the Data Distribution.
>
> [5] Diffusion Models Beat GANs on Image Synthesis.
>
>
>
> > **[Q3] How efficient is Graffe? For example, how does it compare in terms of training time with other contrastive methods or MAE methods?**
>
> Thank you for your constructive feedback. To address your concern, we selected GRACE and GraphMAE as baselines for contrastive and MAE-based methods, respectively. These models are considered classic works in the field, with minimal overengineering and higher efficiency. The runtime statistics were collected on the Cora, Citeseer, and Pubmed datasets. It is important to note that the convergence is challenging to define precisely for self-supervised learning, as the loss typically continues to decrease throughout training. Therefore, we report the runtime per epoch **in terms of second** for comparison. The results are as follows:
>
> | T_epoch (s) | Cora   | Citeseer | Pubmed |
> | ----------- | ------ | -------- | ------ |
> | GRACE       | 0.0317 | 0.0389   | 0.2783 |
> | GraphMAE    | 0.0197 | 0.0254   | 0.0314 |
> | Graffe      | 0.0418 | 0.0675   | 0.1421 |
>
> We observe that for small-scale datasets like Cora and Citeseer, Graffe slightly lag behind GRACE and GraphMAE, but the additional time consumption is acceptable for these datasets considering their scale. For large-scale datasets like Pubmed, while Graffe is slower than GraphMAE, it is two times faster than GRACE, whose time complexity is $\mathcal{O}(N^2)$ when contrasting between node pairs. We would like to emphasize that the efficiency is not the primary focus of our work since it does not introduce significant computational overhead during our experiments. As such, our primary focus remains on the quality of representation learning.

---

> ### Comment · Reviewer_TYZT · 2024-11-26
> **Response to the Author**
>
> Thank you for your detailed response. I have carefully read your response, and am happy with your response regarding the weaknesses. However, I am not quite satisfied with your answers to all three questions, as I feel these responses somewhat evade the main issues and even ignore certain facts.
>
> - Q1: The authors only chose GRACE as the baseline model for comparing the impact of different GNN architectures. This is clearly insufficient, as GRACE is already a method from nearly 4 years ago, and models from three years ago using GCN as the backbone (such as CCA-SSG) have already achieved performance surpassing GRACE, GraphMAE, and Graffe.
>
> - Q2: The experimental results show that incorporating the UNet structure can achieve performance far superior to MLP and GNN. This raises a question: what is the additional parameter count and training load brought by the UNet structure, and is such a comparison fair under these conditions? Moreover, I think comparing epoch-wise training time is very unfair. Different methods have significant differences in convergence speed. For example, GRACE/CCA-SSG can achieve optimal embedding quality in around the first 100 epochs, while BGRL needs thousands of epochs to converge. It would be more reasonable to compare the training time needed to achieve the results shown in Table 1.
>
>
> - Q3: The authors claim that GRACE is "considered classic works in the field, with minimal overengineering and higher efficiency." This assertion is incorrect. GRACE itself requires an additional projector and has O(N^2) complexity, so it clearly doesn't have higher efficiency. At the very least, CCA-SSG has both a simpler architecture and more efficient training than GRACE. Moreover, I think comparing epoch-wise training time is very unfair. Different methods have significant differences in convergence speed. For example, GRACE/CCA-SSG can achieve optimal embedding quality in around the first 100 epochs, while BGRL needs thousands of epochs to converge. It would be more reasonable to compare the training time needed to achieve the results shown in Table 1.
>
> Given the existing experimental results, I am more inclined to attribute Graffe's empirical performance improvements to its more complex encoder/decoder, rather than the introduction of the diffusion model.

---

> > ### Author Response · Authors · 2024-11-28
> > **Further Response to Reviewer TYZT (1/3)**
> >
> > We sincerely thank you for your follow-up constructive questions and apologize for the delay in our response, as we took time to conduct and analyze additional experiments to address the concerns more thoroughly. Below, we provide a point-by-point response.
> >
> > > **[Q1] The authors only chose GRACE as the baseline model for comparing the impact of different GNN architectures. This is clearly insufficient, as GRACE is already a method from nearly 4 years ago, and models from three years ago using GCN as the backbone (such as CCA-SSG) have already achieved performance surpassing GRACE, GraphMAE, and Graffe.**
> >
> > We sincerely thank the reviewer for pointing out this issue. We acknowledge the biased statement regarding Graffe's advantage with the GCN backbone in our initial response and apologize for the oversight. We have revised the relevant description in the manuscript.
> >
> > We want to clarify that **using GAT as the encoder in graph SSL is not universally optimal**. As mentioned in our original response, replacing GCN with GAT often results in performance drops in many SSL methods. To further support this, we extended our analysis by comparing the performance of additional three competitive methods, such as Bandana, MaskGAE (AE branch) and CCA-SSG (contrastive branch), when switching from GCN to GAT on Cora, Citeseer, and PubMed datasets.
> >
> > |                             | Cora             | Citeseer         | PubMed           |
> > | --------------------------- | ---------------- | ---------------- | ---------------- |
> > | **GRACE (GCN)**             | 81.9$\pm$0.4     | 71.2$\pm$0.5     | 80.6$\pm$0.4     |
> > | **GRACE (GAT)**             | 81.0$\pm$0.6     | 71.5$\pm$0.5     | 78.9$\pm$0.2     |
> > | **GraphMAE (GCN)**          | 82.5$\pm$0.5     | 72.6$\pm$0.6     | 80.9$\pm$0.2     |
> > | **GraphMAE (GAT)**          | 84.2$\pm$0.4     | 73.4$\pm$0.4     | 81.1$\pm$0.4     |
> > | **CCA-SSG (GCN)**           | 84.0$\pm$0.4     | 73.1$\pm$0.3     | 81.0$\pm$0.5     |
> > | **CCA-SSG (GAT)**           | 82.7$\pm$0.6     | 72.3$\pm$0.6     | 80.7$\pm$0.9     |
> > | ***MaskGAE$_{edge}$ (GCN)** | 83.8$\pm$0.3     | 72.9$\pm$0.2     | 82.7$\pm$0.3     |
> > | ***MaskGAE$_{edge}$ (GAT)** | 82.0$\pm$0.1     | 72.0$\pm$0.4     | 81.2$\pm$0.1     |
> > | **Bandana (GCN)**           | 84.5$\pm$0.3     | 73.6$\pm$0.2     | **83.7$\pm$0.5** |
> > | **Bandana (GAT)**           | 83.1$\pm$0.6     | 73.7$\pm$0.5     | 81.5$\pm$0.8     |
> > | **Graffe (GCN)**            | 83.2$\pm$0.5     | 73.2$\pm$0.2     | 80.5$\pm$0.4     |
> > | **Graffe (GAT)**            | **84.8$\pm$0.4** | **74.3$\pm$0.4** | 81.0$\pm$0.6     |
> >
> > *(Results marked with \* are taken from the original papers.)*
> >
> > The results show obvious performance declines for many methods when adjusting GCN to GAT, such as CCA-SSG, MaskGAE, and Bandana on Cora and Citeseer, which also aligns with observations in GraphMAE [1] (Appendix A.2 of its literature) and MaskGAE [2] (Table 5 of its literature). Therefore, we believe GSSL methods often favor specific encoders. Thus, **we respectfully disagree with the reviewer’s view that our performance gains are due to the complex encoder choice.**
> >
> > Last but not least, for graph-level experiments, most methods, including ours, use GIN as the encoder. Our performance advantage in Table 2 provides more equitable evidence of the effectiveness of our method.
> >
> > [1] Graphmae: Self-supervised masked graph autoencoders. KDD 2022.
> >
> > [2] What’s Behind the Mask: Understanding Masked Graph Modeling for Graph Autoencoders. KDD 2023.

---

> > ### Author Response · Authors · 2024-11-28
> > **Further Response to Reviewer TYZT (2/3)**
> >
> > > **[Q2] The experimental results show that incorporating the UNet structure can achieve performance far superior to MLP and GNN. This raises a question: what is the additional parameter count and training load brought by the UNet structure, and is such a comparison fair under these conditions?**
> >
> > Regarding the parameter count, the widely adopted GAT encoder contains ~6M parameters, and our Graph-Unet decoder has ~6.2M parameters. In comparison, GraphMAE's single-layer GAT decoder comprises ~2.8M parameters. To respond to questions about the reviewer’s concerns regarding the choice of decoder architecture, especially in terms of parameter count and training load, we would like to emphasize that **a heavier decoder does not generally lead to better performance.**
> >
> > We find relevant discussions on this question from both theoretical and empirical perspectives in the autoencoder setting in [1]. The authors in [1] prove that '*the encoder should be relatively strong to better learn the latent distribution*'. They also argue that *The intuition that (larger encoder, smaller decoder) is preferred to (smaller encoder, larger decoder) for learning a more informative latent is supported by the well-known posterior collapse phenomenon in the VAE literature, where the encoded distribution becomes completely uninformative if the decoder is too powerful.* Furthermore, the experimental results in [1] suggest that a relatively small decoder can encourage the encoder to learn a meaningful latent space. While our setting differs from the traditional autoencoder framework, we believe the underlying intuition is similar, supporting the viewpoint that a more powerful decoder does not necessarily lead to better performance.
> >
> > To validate this in our scenario, we conducted additional experiments where we adjusted the decoders of two classical MAE methods, GraphMAE and MaskGAE, to heavier decoder and Graph-Unet, making their parameter count comparable to our decoder. We carefully adjust hyper-parameters when adopting the modified decoders and report the results of five runs with the best hyperparameter settings:
> >
> > |                                                              | Cora         | Citeseer     | PubMed       |
> > | ------------------------------------------------------------ | ------------ | ------------ | ------------ |
> > | **GraphMAE** (Graph-Unet decoder)                            | 76.9$\pm$0.8 | 71.3$\pm$0.6 | 78.8$\pm$0.3 |
> > | **GraphMAE** (original decoder with comparable #params as ours) | 76.1$\pm$0.8 | 71.5$\pm$0.5 | 79.8$\pm$0.3 |
> > | **GraphMAE** (original)                                      | 84.2$\pm$0.4 | 73.4$\pm$0.4 | 81.1$\pm$0.4 |
> > | **MaskGAE$_{edge}$** (Graph-Unet decoder)                    | 76.8$\pm$0.5 | 70.5$\pm$0.3 | 80.2$\pm$0.3 |
> > | **MaskGAE$_{edge}$** (original decoder with comparable #params as ours) | 75.8$\pm$0.5 | 70.8$\pm$0.5 | 80.5$\pm$0.2 |
> > | **MaskGAE$_{edge}$** (original)                              | 83.8$\pm$0.3 | 72.9$\pm$0.2 | 82.7$\pm$0.3 |
> >
> > The results show that using these modified heavier decoder leads to varying degrees of performance decline for both MAE methods, with up to a drop of 8 percentage point on the Cora dataset.
> >
> > In our initial response, we aimed to convey that Graph-Unet is particularly well-suited for modeling diffusion representation learning. Replacing it with other architectures would limit the expressive power of our method, but this improvement is not attributable to an increase in parameter count. It is also worth noting that previous MAE methods also employed diverse decoder designs. For instance, MaskGAE incorporates two MLP-based decoders: a degree decoder and a structure decoder, while GraphMAE adopt a 1-layer GAT decoder for node-level tasks.
> >
> > Given these observation, using MLPs or GNNs as the decoder would be unfair to our method. Similarly, equipping GraphMAE or MaskGAE with larger decoders would also be unfair, as it would limit their representation learning capabilities and shift the training focus disproportionately toward decoder optimization attributed to its heavy weight.
> >
> > We fully understand the reviewer’s concerns regarding the complexity of our decoder as it indeed. However, we hope the above explanation clarifies that the effectiveness of our method arises from the non-trivial synergy between the decoder and the learning strategy. Therefore, dismissing the validity of our representation learning approach solely based on the decoder’s complexity would be regrettable. Based on the above empirical evidence and previously validated results in [1], **we respectfully disagree with the reviewer’s claim that our performance improvements are attributed to the complex decoder.**
> >
> > [1] Complexity Matters: Rethinking the Latent Space for Generative Modeling. NeurIPS 2023.

---

> > ### Author Response · Authors · 2024-11-28
> > **Further Response to Reviewer TYZT (3/3)**
> >
> > > **[Q3] The authors claim that GRACE is "considered classic works in the field, with minimal overengineering and higher efficiency." This assertion is incorrect. GRACE itself requires an additional projector and has O(N^2) complexity, so it clearly doesn't have higher efficiency. At the very least, CCA-SSG has both a simpler architecture and more efficient training than GRACE. Moreover, I think comparing epoch-wise training time is very unfair. Different methods have significant differences in convergence speed. For example, GRACE/CCA-SSG can achieve optimal embedding quality in around the first 100 epochs, while BGRL needs thousands of epochs to converge. It would be more reasonable to compare the training time needed to achieve the results shown in Table 1.**
> >
> > We appreciate the reviewer's feedback on our previous response. Upon reflection, we acknowledge that  our initial analysis of time efficiency was indeed not sufficiently objective. Following your suggestion, we have added a comparison of **the training time needed to achieve their reported results** for different methods. Additionally, we included a more efficient contrastive learning baseline, CCA-SSG, for comparison. Below are our results, all obtained on an A100 (80GB) GPU:
> >
> > | **T_convergence (s)** | **Cora**            | **Citeseer**       | **PubMed**         |
> > | --------------------- | ------------------- | ------------------ | ------------------ |
> > | **GRACE**             | 4.41(200 epochs)    | 4.66 (200 epochs)  | 41.72 (200 epochs) |
> > | **CCA-SSG**           | 1.46 (20 epochs)    | 1.14 (20 epochs)   | 7.23 (100 epochs)  |
> > | **GraphMAE**          | 19.85 (1500 epochs) | 4.83 (300 epochs)  | 36.41(1000 epochs) |
> > | **Graffe**            | 22.77 (800 epochs)  | 12.02 (200 epochs) | 33.89 (300 epochs) |
> >
> > As observed, CCA-SSG is the most time-efficient method across all three datasets, requiring the fewest epochs to converge. While GraphMAE has the shortest per-epoch training time, its overall time is higher due to the larger number of epochs needed for convergence. GRACE performs well on smaller graphs but takes the longest time on large-scale graph like PubMed. Our method has the longest training time on Cora and Citeseer but slightly outperforms GraphMAE and GRACE on large-scale graph PubMed.
> >
> > In summary, Graffe does not have an general advantage in terms of time efficiency. However, as stated in our initial response, *efficiency is not the primary focus of our work* as it does not result in heavy training burdens. Our goal is to introduce a novel representation learning method that may inspire the community with a new learning paradigm. Improvements in training speed and further refinements in the design are left as directions for future research. We hope the above analysis addresses the reviewer’s concerns regarding training time.

---

> > ### Author Response · Authors · 2024-12-01
> >
> > Dear Reviewer TYZT,
> >
> > Thanks for your contributions to the reviewing process. As the deadline for the discussion approaches, we kindly request your feedback on whether our further response has addressed your concerns. We would be more than willing to engage in further discussions and make any necessary improvements.
> >
> > Thank you for your time and consideration.
> >
> > Best wishes,
> >
> > Authors

---

> > > ### Comment · Reviewer_TYZT · 2024-12-01
> > >
> > > Thank you for the further clarification and additional results. I will maintain the positive rating.

---

### Official Review · Reviewer_kLs2 · 2024-11-07

**Soundness:** 3
**Presentation:** 3
**Contribution:** 3
**Rating:** 6
**Confidence:** 2

**Summary:**

In this submission, the authors propose a self-supervised diffusion model, called Graffe, for graph representation learning.
Graffe generates a source graph representation via a trainable encoder and leverages the representation as the condition of the diffusion model.
In particular, the authors prove that the denoising objective implicitly maximizes the conditional mutual information between data and its representation, and the negative logarithm of denoising score matching loss is a tractable lower bound for the conditional mutual information.
Experiments on real-world datasets further verify the feasibility of the proposed method.

**Strengths:**

- Although there have been a few works on leveraging diffusion models for graph representation learning[1,4] or considering the representation abilities of DPM[3], I find this submission quite interesting. The rationale is convincing, and it thoroughly discusses the design of the proposed method.
- The theoretical part seems correct and reveals some interesting results that further support the proposed method's feasibility in practice.

**Weaknesses:**

- The experimental results for node classification are promising. However, in the graph classification task, it appears that the SOTA method is GIP[2], which significantly enhances performance in this area. It would be beneficial for the authors to compare Graffe with GIP.

[1] Xian, Jia Jun Cheng, Sadegh Mahdavi, Renjie Liao, and Oliver Schulte. "From Graph Diffusion to Graph Classification." In ICML 2024 Workshop on Structured Probabilistic Inference {\&} Generative Modeling.

[2] Zhao, Xinjian, Wei Pang, Xiangru Jian, Yaoyao Xu, Chaolong Ying, and Tianshu Yu. "Enhancing Graph Self-Supervised Learning with Graph Interplay." arXiv preprint arXiv:2410.04061 (2024).

[3] Komanduri, Aneesh, Chen Zhao, Feng Chen, and Xintao Wu. "Causal Diffusion Autoencoders: Toward Representation-Enabled Counterfactual Generation via Diffusion Probabilistic Models."

[4] Yang, Run, Yuling Yang, Fan Zhou, and Qiang Sun. "Directional diffusion models for graph representation learning." Advances in Neural Information Processing Systems 36 (2024).

**Questions:**

Please see above.

---

> ### Author Response · Authors · 2024-11-20
> **Response to Reviewer kLs2**
>
> > **1. The experimental results for node classification are promising. However, in the graph classification task, it appears that the SOTA method is GIP[2], which significantly enhances performance in this area. It would be beneficial for the authors to compare Graffe with GIP.**
>
> Thank you for bringing this work to our attention. We noticed that this paper is a preprint published on Arxiv in October 2024, which is after the submission deadline for this year’s ICLR. Nevertheless, we are happy to discuss this work with the reviewer. After carefully reading the paper, we identified two significant differences between GIP and Graffe, which make it difficult to directly compare our method with GIP. Below is a detailed analysis:
>
> - **Different Research Motivations**: Inspired by graph-level communications, GIP was proposed as a plug-and-play method to enhance existing graph self-supervised learning approaches by integrating random inter-graph edges within batches. In other words, GIP itself is not a self-supervised learning method but rather a data augmentation approach similar to DropEdge [1], as evidenced in Table 1 of its paper. Furthermore, it is worth noting that GIP cannot be applied to node-level tasks. On the other hand, our method is a novel graph self-supervised learning framework that focuses on designing training objectives to improve the quality of representations for both node-level and graph-level tasks. Therefore, the research **motivations of these two works are fundamentally different**.
>
> - **Incompatibility with Non-Contrastive Frameworks**: The data augmentation approach of GIP is exclusively suited for contrastive learning frameworks. As analyzed in Theorem 1 of its paper, manifold separation is achieved through multi-view contrastive learning. Unfortunately, this **incompatibility** prevents us from incorporating GIP into the Graffe framework to report experimental results.
>
> Interestingly, while GIP combined with contrastive learning shows impressive performance on TU datasets, its performance on OGB datasets is less promising. We hypothesize that this could be due to GIP’s effectiveness being highly dependent on the connectivity of graphs. We hope our response about GIP addresses your concern.
>
> [1] DropEdge: Towards Deep Graph Convolutional Networks on Node Classification. ICLR 2020.

---

### Author Response · Authors · 2024-11-20
**General Response**

### **General Response**
We would like to thank all the reviewers very much for their extensive reviews and constructive critiques. We are encouraged that reviewers find that our approach combines graph SSL with diffusion models in a promising and exciting way (Reviewer TYZT), that the experiments are comprehensive and support the effectiveness (Reviewer TYZT, W23D, and 36MZ), that the rationale is convincing and with a necessary theoretical foundation (Reviewers kLs2 and 36MZ).

However, we noticed that the research problem targeted in our study has not been fully captured, leading to potential misunderstandings among some reviewers. Therefore, we would like to **restate our research problem** to address related concerns:

- **Research Problem**: Since graph representation learning encompasses several subdomains, we wish to emphasize that our method focuses on designing efficient **learning strategies** rather than developing more expressive **GNN backbones**. Specifically, our goal is to design a self-supervised optimization objective that incorporates denoising score matching loss to enhance the quality of the resulting node/graph representations. Therefore, our approach is orthogonal to those methods focusing on GNN architecture design, representing distinct directions in the development of graph representation learning.


### **Content of Revisions**

Moreover, in response to the reviewers' suggestions, we have made appropriate revisions to the original manuscript to better address these concerns. Below, we provide **a brief summary** of the revisions for the reviewers' reference:

- We have conducted the ablation study on encoder and decoder design in `Appendix D`. (for reviewer TYZT Quesion 1and 2.)
- We have supplemented discussions about intuitive guide for choosing masking ratio and the potential of structure information in DRL in `Appendix E`. (for reviewer W23D Weakness 1 and 2.)
- We have revised the description in the `Introduction` (for Reviewer W23D Weakness 2) and notations in `Section 2.1` and `Theorem 1`. (for reviewer 36MZ Weakness 2.)

Finally, we appreciate all your helpful comments that strengthen the quality and clarity of our work. we hope the following responses address your concerns and we look forward to engaging in an active and productive discussion with the reviewers. If you have any other questions, please feel free to let us know.

---

### Meta-Review · Area_Chair_HiBP · 2024-12-18

**Metareview:**

In this submission, the authors proposed a diffusion probabilistic model for graph representation, in which a graph encoder is learned to provide informative conditions to guide the denoising (inverse diffusion) process. Although all reviewers scored the submission on the positive borderline. Some critical concerns are unresolved:

1) As claimed by the authors, the proposed method implicitly achieves mutual information maximization, and the score matching loss corresponds to the lower bound of the conditional mutual information. Because this principle has been widely used by other graph representation learning methods, especially those contrastive learning methods, it is unknown why the proposed method can work better (considering that the proposed method merely optimizes the lower bound of the original mutual information loss).

2) In the aspect of experimental results, the improvements caused by the proposed method are incremental to some degree.

Therefore, I think the submission requires a next-round review.

**Additional Comments On Reviewer Discussion:**

In the rebuttal phase, three of the four reviewers interacted with the authors, and some of them were satisfied with the rebuttals and thus maintained or increased their scores. In the discussion and decision phases, AC had more concerns about the rationality of the proposed method and asked for more discussions. Reviewer TYZT provided a new comment to support AC's opinion --- both AC and Reviewer TYZT think that the performance improvement is from the modification of the model architecture rather than from the proposed loss function, which makes the contribution and the motivation of the proposed method questionable. After reading the submission and the comments, AC tends to reject this paper.

---

### Decision · Program_Chairs · 2025-01-22

Reject